# Towards Privacy-Guaranteed Label Unlearning in Vertical Federated Learning: Few-Shot Forgetting without Disclosure

**Hanlin Gu**[*]
WeBank AI Lab

**Hong Xi Tae**[*]
Universiti Malaya

**Lixin Fan**
WeBank AI Lab

**Chee Seng Chan**[†]
Universiti Malaya

## Abstract

This paper addresses the critical challenge of unlearning in Vertical Federated Learning (VFL), a setting that has received far less attention than its horizontal counterpart. Specifically, we propose the first method tailored to *label unlearning* in VFL, where labels play a dual role as both essential inputs and sensitive information. To this end, we employ a representation-level manifold mixup mechanism to generate synthetic embeddings for both unlearned and retained samples. This is to provide richer signals for the subsequent gradient-based label forgetting and recovery steps. These augmented embeddings are then subjected to gradient-based label forgetting, effectively removing the associated label information from the model. To recover performance on the retained data, we introduce a recovery-phase optimization step that refines the remaining embeddings. This design achieves effective label unlearning while maintaining computational efficiency. We validate our method through extensive experiments on diverse datasets, including MNIST, CIFAR-10, CIFAR-100, ModelNet, Brain Tumor MRI, COVID-19 Radiography, and Yahoo Answers demonstrate strong efficacy and scalability. Overall, this work establishes a new direction for unlearning in VFL, showing that re-imagining mixup as an efficient mechanism can unlock practical and utility-preserving unlearning. The code is publicly available at https://github.com/bryanhx/Towards-Privacy-Guaranteed-Label-Unlearning-in-Vertical-Federated-Learning.

## 1 Introduction

Vertical Federated Learning (VFL) (Yang et al., 2019) enables multiple organizations to collaboratively utilize their private datasets in a privacy-preserving manner, even when they share some sample IDs but differ significantly in terms of features. In VFL, there are typically two types of parties: (i) the passive party, which holds the *features*, and (ii) the active party, which possesses the *labels*. VFL has seen the widespread application, especially in sensitive domains like banking and healthcare, where organizations benefit from joint modeling without exposing their raw data (Li et al., 2020).

A fundamental requirement in VFL is the necessity for unlearning, driven by the *"right to be forgotten"* as mandated by regulations such as GDPR[1] and CCPA[2]. While unlearning has been explored in Horizontal Federated Learning (HFL), there has been limited attention to its application in vertical settings. Existing studies on vertical federated unlearning (Li et al., 2024; Wang et al., 2024; Han et al., 2025) primarily address the removal of features when an entire passive client withdraws. In contrast, **this paper focuses on label unlearning**, which is particularly critical in scenarios where labels encode highly sensitive information. For example, in medical diagnostics, one may wish to *erase the label indicating whether a patient is HIV-positive*, as it reveals private health status. Similarly, in credit risk assessment systems, *revoking the label associated with a loan approval decision* helps safeguard against fairness and regulatory concerns.

---

[*]These authors contributed equally to this work; authors are listed alphabetically by first name.

[†]Corresponding author: Chee Seng Chan (cs.chan@um.edu.my).

[1]https://gdpr-info.eu/art-17-gdpr/

[2]https://oag.ca.gov/privacy/ccpa

However, the need for synchronous processing is a paramount constraint in VFL architectures. In VFL, different parties possess unique feature sets over a common sample ID space, necessitating coordinated training steps in which intermediate results are exchanged and aligned at precise points. This rigorous coordination implies that all participating models must wait for the slowest participant to complete its step before proceeding to the next round. Consequently, this synchronization requirement significantly compounds the difficulties associated with unlearning efficiency in the context of Vertical Federated Unlearning (VFU).

To overcome this challenge, we introduce a novel few-shot unlearning framework for VFL that removes labels from both active and passive models, relying solely on a small public dataset. To achieve this, we re-purpose the manifold mixup (Verma et al., 2019) technique, typically used for data augmentation to generate synthetic embeddings for both unlearned and retained samples. This approach accelerates unlearning efficiency while maintaining good unlearning performance even with a limited number of data samples. On these transformed embeddings, the active party performs gradient-based forgetting, while inverse gradients allow the passive party to unlearn the corresponding representations locally without accessing raw labels. A final recovery phase restores accuracy on the retained data.

Our proposed unlearning solution offers three key advantages: (i) **Performance-Preserving Unlearning**: It requires only a small subset of labeled public data, significantly minimizing the degradation of the retain classes performance; (ii) **Enhanced Unlearning Effectiveness**: By leveraging the manifold mixup technique, it achieves effective unlearning with minimal data; and (iii) **Computational Efficiency**: The proposed unlearning process is highly efficient, completing within seconds.

The primary contributions of this work are as follows:

i. To the best knowledge, this is the first work to address label unlearning in VFL.

ii. We propose a novel few-shot label unlearning method that removes labels from both active and passive models in VFL using only a limited amount of public data. As a result, our approach enhances unlearning effectiveness through a representation-level manifold mixup mechanism, ensuring a more robust and efficient unlearning process (see Section 4).

iii. We conduct extensive experiments on diverse benchmark datasets, including MNIST (image), CIFAR-10 (image), CIFAR-100 (image), ModelNet (image), Brain Tumor MRI (image), COVID-19 Radiography (image) and Yahoo Answers (text). The results demonstrate that our approach rapidly and effectively unlearns target labels, outperforming existing machine unlearning techniques in both efficiency and effectiveness.

## 2 RELATED WORKS

This section reviews related work in two areas, (i) machine unlearning (MU) and (ii) federated unlearning (FU). Please see Appendix A.7 for the comparison of our method with existing FU studies.

### 2.1 MACHINE UNLEARNING (MU)

MU refers to the process of selectively removing the influences of specific data points from a machine learning (ML) model after it has been trained. This process ensures that the model "forgets" the contributions of specific data, as though the data had never been part of the training process.

MU was first proposed by (Cao & Yang, 2015) to remove specific data from a model without full retraining (Garg et al., 2020; Chen et al., 2021). It includes exact and approximate unlearning. Exact methods like SISA (Bourtoule et al., 2021) and ARCANE (Yan et al., 2022) partition data, train sub-models per partition, and retrain only the affected sections during unlearning. Approximate unlearning includes methods like fine-tuning (Golatkar et al., 2020a; Jia et al., 2023), random label updates (Graves et al., 2021; Chen et al., 2023), noise injection (Tarun et al., 2024; Huang et al., 2021), gradient ascent (Goel et al., 2022; Choi & Na, 2023; Abbasi et al., 2023; Hoang et al., 2024), knowledge distillation (Chundawat et al., 2023b; Zhang et al., 2023b; Kurmanji et al., 2023), and weight scrubbing (Golatkar et al., 2020a;b; 2021; Guo et al., 2020; Foster et al., 2024a). While these rely on data during unlearning, Chundawat et al. (Chundawat et al., 2023a) propose zero-shot unlearning, and Yoon et al. (Yoon et al., 2022) introduce a few-shot approach using model inversion.

## 2.2 Federated Unlearning (FU)

In FU, most of the existing works are focusing in the horizontal environment (Wu et al., 2022; Gu et al., 2024a; Zhao et al., 2023; Romandini et al., 2024; Liu et al., 2024b; Zhang et al., 2023a; Su & Li, 2023; Ye et al., 2024; Gao et al., 2024; Cao et al., 2023b; Yuan et al., 2023; Alam et al., 2024; Li et al., 2023; Halimi et al., 2022; Xia et al., 2023; Wang et al., 2023; Dhasade et al., 2023; Liu et al., 2022; Zhao et al., 2024; Wang et al., 2022; Gu et al., 2024b; Liu et al., 2021; Cao et al., 2023a). The research on horizontal federated unlearning (HFU) mainly focuses on label (Wang et al., 2022; Zhao et al., 2024), client (Liu et al., 2021; Yuan et al., 2023; Zhang et al., 2023a; Gao et al., 2024; Ye et al., 2024; Su & Li, 2023; Cao et al., 2023a; Wu et al., 2022) and sample unlearning (Liu et al., 2022; Dhasade et al., 2023). While HFU has been widely studied, vertical federated unlearning (VFU) remains underexplored. Existing works mainly focus on passive party unlearning, such as logistic regression (Deng et al., 2023), gradient boosting (Li et al., 2024), and deep learning models with fast retraining (Wang et al., 2024) or backdoor defence (Han et al., 2025). However, little attention has been given to active-party-initiated unlearning with all parties actively engaged in VFL.

## 3 Background

This section introduces the general setup of the label unlearning process.

### 3.1 General Setup

**VFL training.** Vertical Federated Learning (VFL) allows parties with different user attributes to collaboratively train a model without sharing raw data (Yang et al., 2019; Liu et al., 2024a). The active party holds the labels and the active model, while passive parties provide features and passive models, enabling performance gains while preserving privacy. We assume that a VFL setting consists of one active party $P_0$ and $K$ passive parties $\{P_1, \cdots, P_K\}$ who collaboratively train a VFL model $\Theta = (\theta_1, \cdots, \theta_K, \omega)$ to optimize:

$$\min_{\omega, \theta_1, \cdots, \theta_K} \frac{1}{n} \sum_{i=1}^{n} \ell(F_\omega \circ (G_{\theta_1}(x_{1,i}), G_{\theta_2}(x_{2,i}), \cdots, G_{\theta_K}(x_{K,i})), y_i), \quad (1)$$

We denote the empirical training objective in Equation 1 as

$$\mathcal{L}(\omega, \theta_1, \ldots, \theta_K) = \frac{1}{n} \sum_{i=1}^{n} \ell(F_\omega \circ (G_{\theta_1}(x_{1,i}), \ldots, G_{\theta_K}(x_{K,i})), y_i).$$

where passive party $P_k$ owns features $x_k = (x_{k,1}, \cdots, x_{k,n})$ and the passive model $G_{\theta_k}$; while the active party owns the labels $y = \{y_1, \cdots, y_n\}$ and active model $F_\omega$. Before training, both parties use Private Set Intersection (PSI) to align data records by the same ID. During training, each passive party $k$ computes a forward embedding $H_k$ from its features and sends it to the active party. The active party concatenates embeddings $\{H_k\}_{k=1}^{K}$ into $H$, generates outputs, computes $\mathcal{L}$, and updates its model using $\frac{\partial \mathcal{L}}{\partial \omega}$. It then sends $\frac{\partial \mathcal{L}}{\partial H_k}$ to passive parties, which compute $\frac{\partial \mathcal{L}}{\partial \theta_k} = \frac{\partial \mathcal{L}}{\partial H_k} \cdot \frac{\partial H_k}{\partial \theta_k}$ to update their models. Please refer to Appendix A.2 for the table of notations.

**Label unlearning in VFL.** When the active party requests unlearning for a subset of samples with indices $\mathcal{I}^u \subseteq \{1, \ldots, n\}$, we define the unlearned dataset as $\mathcal{D}^u = \{(x_{1,i}, \ldots, x_{K,i}, y_i)\}_{i \in \mathcal{I}^u}$. For each passive party $P_k$, the corresponding feature subset is $x_k^u = (x_{k,i})_{i \in \mathcal{I}^u}$.

The objective of label unlearning is to efficiently remove the influence of $\mathcal{D}^u$ from the trained VFL system without requiring full retraining. Formally, given an unlearning mechanism $\mathcal{U}$ acting on the trained model $\Theta = (\theta_1, \ldots, \theta_K, \omega)$, the updated parameters are

$$(\theta_1^u, \ldots, \theta_K^u, \omega^u) = \mathcal{U}(\Theta; \mathcal{D}^u).$$

Following standard machine unlearning principles (Bourtoule et al., 2021), an effective unlearning method should satisfy: i) **Performance-Preserving Unlearning**, removing the impact of $\mathcal{D}^u$ without degrading performance on retained data; ii) **Enhanced Unlearning Effectiveness**, requiring minimal auxiliary data; and iii) **Computational Efficiency**, avoiding full model retraining.

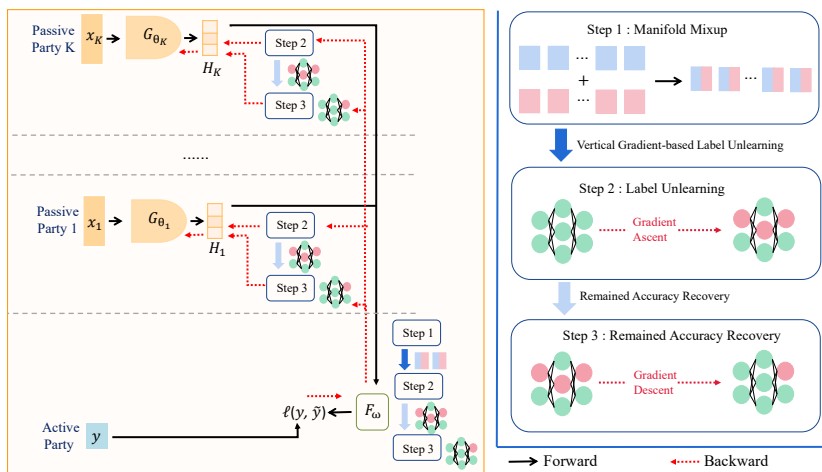

Figure 1: Overview of our proposed few-shot unlearning framework in VFL setting.

# 4 FEW-SHOT LABEL UNLEARNING

This section outlines our proposed few-shot label unlearning method, as shown in Figure 1 and Algorithm 1. The approach involves three steps: (1) augmenting the forward embedding with manifold mixup to address limited labeled data (Section 4.1); (2) using gradient ascent on the augmented embedding to guide label removal in both passive and active models (Section 4.2); and (3) improving model accuracy on the retained labels (Section 4.3).

## 4.1 VERTICAL MANIFOLD MIXUP

With fewer samples, the steps involved in the unlearning algorithm, such as forward embeddings execution and gradient updates, can be executed much faster. However, this small labeled unlearning set, denoted as $\mathcal{D}^{p,u} = \{(\{x_{k,i}^{p,u}\}_{k=1}^K, y_i^{p,u})\}_{i=1}^{n_{p,u}}$ is often insufficient for effective unlearning (see Figure 3). Consequently, we cast this setting as a *few-shot unlearning* problem, where only a small number of labeled unlearning samples are available to remove the corresponding label information from the model. Please refer to Appendix A.4 for an explanation of why a small number of samples (up to 40 per label, as shown in Tables 9 and 10) is sufficient to achieve performance comparable to using the full dataset.

Drawing inspiration from the principles of few-shot learning, we repurpose manifold mixup (Verma et al., 2019) by interpolating hidden embeddings rather than directly mixing features. We propose a manifold mixup framework for VFL by optimizing the following loss function:

$$\max_{\omega,\theta_1,\ldots,\theta_K} \frac{1}{n_{p,u}^2} \sum_{i,j=1}^{n_{p,u}} \ell\Big( F_\omega\Big(\text{Mix}_\lambda(G_{\theta_1}(x_{1,i}^{p,u}), G_{\theta_1}(x_{1,j}^{p,u})), \ldots, \text{Mix}_\lambda(G_{\theta_K}(x_{K,i}^{p,u}), G_{\theta_K}(x_{K,j}^{p,u}))\Big), \text{Mix}_\lambda(y_i^{p,u}, y_j^{p,u})\Big).$$

where

$$\text{Mix}_\lambda(a, b) = \lambda \cdot a + (1 - \lambda) \cdot b. \tag{2}$$

The mixed coefficient $\lambda$ ranges from 0 to 1. The advantage of the manifold mixup approach lies in its ability to flatten the state distributions (Verma et al., 2019). Once each passive party $k$ generates its local embeddings $H_{k,i}^{p,u} = G_{\theta_k}(x_{k,i}^{p,u})$, these embeddings are transmitted to the active party. The active party then constructs a set of synthesized embeddings $\vec{H}_k^u$ by performing manifold mixup among embeddings originating from the same passive party, ensuring consistency in the $\lambda$ value while avoiding any coordination among passive parties. Similarly, for the small data with the remaining label $\mathcal{D}^{p,r}$, we also implement the manifold mixup to obtain the $\vec{H}_k^r$ and corresponding label $\vec{y}^r$ for the remaining accuracy recovery step.

## 4.2 VERTICAL GRADIENT-BASED LABEL UNLEARNING

Once the augmented embeddings $\{\vec{H}_1, \ldots, \vec{H}_K^u\}$ are generated, a straightforward yet effective strategy is to perform gradient ascent on both the active and passive models using these augmented embeddings. Specifically, the active party concatenates all embeddings $\{\vec{H}_k^u\}_{k=1}^K$ into a single tensor $\vec{H}^u = [\vec{H}_1^u, \ldots, \vec{H}_K^u]$, and optimizes it according to the following formulation:

$$\max_\omega \ell(F_\omega(\vec{H}^u), \vec{y}^u) = \ell(F_\omega([\vec{H}_1, \cdots, \vec{H}_K^u]), \vec{y}^u), \tag{3}$$

where $\vec{y}^u$ represents the mixture of the representative unlearned labels and $\eta$ is the learning rate.

**Unlearning for active model $F_\omega$.** On one hand, the active model undergoes unlearning for the active model $F_\omega$ via gradient ascent as follows:

$$\omega = \omega + \eta \nabla_\omega \ell(F_\omega(\vec{H}^u), \vec{y}^u). \tag{4}$$

**Unlearning for passive model $G_{\theta_k}$.** Subsequently, the active party computes the gradients $\frac{\partial \ell}{\partial \vec{H}_k^u}$ in accordance with Equation 3 and transmits these gradients to the corresponding passive party $k$. Finally, the passive party $k$ updates the passive model $G_{\theta_k}$ using the following expression:

$$\theta_k = \theta_k + \eta \nabla_{\vec{H}^u} \ell(F_\omega(\vec{H}^u), \vec{y}^u) \cdot \nabla_{\theta_k} \vec{H}_k^u. \tag{5}$$

**Theorem 1.** *Suppose that both the trained passive model $\theta$ and the active model $\omega$ achieve a training loss bounded by a small value $\epsilon$. Then, when unlearning a single label, the following holds:*

$$\mathbb{E}_{(\vec{H}^u, \vec{y}^u)} \nabla_\omega \ell(\omega; \vec{H}^u, \vec{y}^u) \cdot \mathbb{E}_{(H^u, y^u)} \nabla_\omega \ell(\omega; H^u, y^u) > 0,$$
$$\mathbb{E}_{(\vec{H}^u, \vec{y}^u)} \nabla_\theta \ell(\theta; \vec{H}^u, \vec{y}^u) \cdot \mathbb{E}_{(H^u, y^u)} \nabla_\theta \ell(\theta; H^u, y^u) > 0, \tag{6}$$

*where $(\vec{H}^u, \vec{y}^u)$ denotes the manifold mixup embeddings and labels of the public data $\mathcal{D}^{p,u}$ associated with the unlearned label, $(H^u, y^u)$ denotes the embeddings and labels of the complete unlearned dataset $\mathcal{D}^u$., and $\ell$ is the main task loss of VFL.*

Theorem 1 indicates that the gradient update direction for label unlearning, when applied to the augmented embeddings of the public unlearned data, is positively aligned with the update direction derived from the embeddings of the entire unlearned dataset. This result suggests that gradient-based label unlearning using only public data is effective and approximates the behavior of unlearning with access to all unlearned data. See proof in Appendix A.1.

## 4.3 REMAINED ACCURACY RECOVERY

The preceding step focuses solely on unlearning the target labels and does not explicitly account for the model's accuracy on the retained data. As a result,

---

**Algorithm 1** Our proposed unlearning

1: **Input:** Bottom models parameters $\theta_k$ of $K$ passive parties, top model parameters $\omega$, unlearn data $\mathcal{D}^{p,u}$, remain data $\mathcal{D}^{p,r}$, learning rate $\eta$, unlearn epoch $N$, batch size $b$.
2: **Output:** Unlearned bottom models parameters $\theta_k^u$, unlearned top model parameters $\omega^u$
3: **for** $n$ in $N$ **do**
4:     **for** $k = 1$ to $K$ **do**
5:         Each passive party $k$ generates local embeddings and send it to *Active party*.
6:     **end for**
7:     ▷ *Manifold Mixup*
8:     Active party generate $\vec{H}_k^u$ and $\vec{H}_k^r$ according to Equation 2.
9:     ▷ *Gradient-based Label Unlearning*
10:     Active party updates $\omega$ according to Equation 4
11:     Active party transfers $\frac{\partial \ell}{\partial \vec{H}_k^u}$ to all passive parties.
12:     **for** $k = 1$ to $K$ **do**
13:         Each Passive party $k$ update $\theta_k$ as Equation 5.
14:     **end for**
15:     ▷ *Remained Accuracy Recovery*
16:     Active party updates $\omega$ according to Equation 7
17:     Active party transfers $\frac{\partial \ell}{\partial \vec{H}_k^u}$ to all passive parties.
18:     **for** $k = 1$ to $K$ **do**
19:         Each Passive party $k$ update $\theta_k$ as Equation 7.
20:     **end for**
21: **end for**
22: **Return** unlearned passive model $\theta_k \triangleq \theta_k^u$ and unlearned active model $\omega \triangleq \omega^u$.

---

the predictive performance on the remaining dataset may degrade. To address this issue, we introduce a Remained Accuracy Recovery step to enhance model accuracy on the retained samples. Specifically, using the small dataset with retained labels, denoted as $\mathcal{D}^{p,r}$, we jointly optimize the passive and active models with respect to the main task loss, formulated as follows:

$$\omega = \omega - \eta \nabla_\omega \ell(F_\omega(\vec{H}^r), \vec{y}^r),$$
$$\theta_k = \theta_k - \eta \nabla_{\vec{H}^r} \ell(F_\omega(\vec{H}^r), \vec{y}^r) \cdot \nabla_{\theta_k} \vec{H}_k^r. \tag{7}$$

| Model | Datasets | Metrics | Accuracy (%) | | | | | | | | |
|---|---|---|---|---|---|---|---|---|---|---|---|
| | | | Baseline | Retrain | FT | Fisher | Amnesiac | Unsir | BU | SSD | Ours |
| ResNet18 | MNIST | $\mathcal{D}^r$ | 99.29 | 99.33 ± 0.03 | 98.99 ± 0.05 | 12.16 ± 0.46 | 98.16 ± 0.92 | 84.92 ± 1.13 | 98.72 ± 0.02 | 96.50 ± 0.19 | **99.05 ± 0.01** |
| | | $y^u$ | 99.39 | 0.00 ± 0.00 | **0.00 ± 0.00** | **0.00 ± 0.00** | **0.00 ± 0.00** | **0.00 ± 0.00** | 58.83 ± 1.79 | **0.00 ± 0.00** | **0.00 ± 0.00** |
| | | ASR | 90.61 | 1.03 ± 0.24 | 2.92 ± 1.08 | **0.11 ± 0.07** | **0.00 ± 0.00** | 29.07 ± 7.95 | 0.47 ± 0.01 | **0.00 ± 0.00** | 0.35 ± 0.01 |
| | CIFAR10 | $\mathcal{D}^r$ | 90.61 | 91.26 ± 0.12 | 88.16 ± 0.15 | 54.4 ± 10.77 | 86.37 ± 0.20 | 75.02 ± 1.65 | 72.65 ± 0.55 | 87.17 ± 0.76 | **89.29 ± 0.19** |
| | | $y^u$ | 93.10 | 0.00 ± 0.00 | 11.00 ± 0.10 | **0.00 ± 0.00** | **0.00 ± 0.00** | **0.00 ± 0.00** | 3.25 ± 0.15 | **0.00 ± 0.00** | **0.00 ± 0.00** |
| | | ASR | 83.84 | 25.98 ± 1.27 | **15.85 ± 2.33** | 50.67 ± 12.51 | **1.62 ± 0.54** | 76.78 ± 0.44 | 34.90 ± 1.16 | 25.74 ± 2.78 | 17.23 ± 1.00 |
| | CIFAR100 | $\mathcal{D}^r$ | 71.43 | 71.03 ± 0.12 | 66.86 ± 0.73 | 61.04 ± 8.61 | 60.05 ± 0.03 | 59.32 ± 0.14 | 55.30 ± 0.81 | 67.25 ± 0.91 | **69.96 ± 0.12** |
| | | $y^u$ | 83.00 | 0.00 ± 0.00 | 12.25 ± 2.25 | **0.00 ± 0.00** | **0.00 ± 0.00** | **0.00 ± 0.00** | 3.50 ± 0.50 | **0.00 ± 0.00** | **0.00 ± 0.00** |
| | | ASR | 88.40 | 25.53 ± 3.36 | 29.30 ± 2.70 | 28.10 ± 4.10 | **2.60 ± 1.30** | 73.70 ± 1.70 | **6.00 ± 0.60** | 6.80 ± 0.01 | 5.02 ± 1.01 |
| | ModelNet | $\mathcal{D}^r$ | 94.26 | 93.90 ± 0.11 | 66.64 ± 1.53 | 28.10 ± 0.69 | 73.91 ± 1.83 | 13.51 ± 0.05 | 24.07 ± 0.27 | 81.89 ± 0.98 | **87.69 ± 0.21** |
| | | $y^u$ | 100.00 | 0.00 ± 0.00 | **0.00 ± 0.00** | **0.00 ± 0.00** | **0.00 ± 0.00** | **0.00 ± 0.00** | **0.00 ± 0.00** | 4.79 ± 1.02 | 2.00 ± 0.00 |
| | | ASR | 98.40 | 0.65 ± 0.05 | 0.79 ± 0.16 | 23.48 ± 0.77 | 1.11 ± 0.16 | 49.20 ± 1.25 | 21.16 ± 0.23 | 0.77 ± 0.03 | **0.11 ± 0.03** |
| | Brain MRI | $\mathcal{D}^r$ | 97.46 | 98.81 ± 0.34 | 81.89 ± 0.82 | 30.26 ± 0.21 | 78.29 ± 0.09 | 73.29 ± 0.09 | 64.19 ± 0.91 | 85.93 ± 0.09 | **93.79 ± 0.17** |
| | | $y^u$ | 97.29 | 0.00 ± 0.00 | 4.33 ± 0.49 | **0.00 ± 0.00** | 3.67 ± 0.14 | **0.00 ± 0.00** | 1.00 ± 0.92 | | 0.56 ± 0.18 |
| | | ASR | 82.32 | 48.37 ± 1.31 | **24.07 ± 0.16** | 51.77 ± 0.83 | 24.29 ± 0.91 | 27.32 ± 3.04 | 58.36 ± 0.92 | 52.35 ± 0.76 | 45.71 ± 0.99 |
| | COVID-19 Radiography | $\mathcal{D}^r$ | 92.82 | 93.85 ± 0.12 | 73.84 ± 0.73 | 40.98 ± 4.89 | 81.76 ± 0.77 | 72.30 ± 0.37 | 66.13 ± 0.69 | 81.11 ± 0.15 | **92.35 ± 0.77** |
| | | $y^u$ | 73.33 | 0.00 ± 0.00 | **0.00 ± 0.00** | **0.00 ± 0.00** | **0.00 ± 0.00** | 10.83 ± 1.48 | 6.11 ± 1.92 | **0.00 ± 0.00** | **0.00 ± 0.00** |
| | | ASR | 88.97 | 46.67 ± 2.74 | **20.51 ± 2.28** | 49.27 ± 0.99 | 13.16 ± 3.92 | 37.52 ± 5.32 | 18.12 ± 2.63 | 36.17 ± 0.02 | 39.21 ± 0.11 |
| Vgg16 | CIFAR10 | $\mathcal{D}^r$ | 89.50 | 90.27 ± 0.19 | 88.69 ± 0.08 | 15.93 ± 4.82 | 84.67 ± 0.22 | 74.74 ± 0.72 | 82.69 ± 0.1 | 87.19 ± 0.39 | **88.97 ± 0.18** |
| | | $y^u$ | 91.10 | 0.00 ± 0.00 | 4.25 ± 1.05 | **0.00 ± 0.00** | **0.00 ± 0.00** | **0.00 ± 0.00** | 2.85 ± 0.05 | **0.00 ± 0.00** | 0.13 ± 0.04 |
| | | ASR | 81.66 | 33.10 ± 1.86 | **21.84 ± 2.66** | 42.25 ± 6.23 | **2.36 ± 0.86** | 21.75 ± 2.41 | 34.53 ± 0.65 | 29.19 ± 0.01 | 31.33 ± 0.36 |
| | CIFAR100 | $\mathcal{D}^r$ | 65.48 | 65.32 ± 0.32 | 59.92 ± 0.56 | 35.42 ± 1.95 | 55.83 ± 0.13 | 55.78 ± 0.59 | 52.21 ± 0.00 | 64.11 ± 0.73 | **64.24 ± 0.09** |
| | | $y^u$ | 77.00 | 0.00 ± 0.00 | 2.50 ± 0.25 | **0.00 ± 0.00** | **0.00 ± 0.00** | **0.00 ± 0.00** | 3.00 ± 0.00 | 1.19 ± 0.03 | 1.00 ± 0.00 |
| | | ASR | 87.20 | 42.13 ± 2.73 | **34.50 ± 4.30** | 40.70 ± 3.50 | **3.10 ± 0.15** | 42.70 ± 0.70 | 18.20 ± 0.11 | 39.11 ± 0.91 | 20.28 ± 0.67 |

Table 1: Accuracy of $\mathcal{D}^r$, $y^u$ and ASR for each unlearning method across ResNet18 and Vgg16 models in single-label unlearning.

## 5 EXPERIMENTAL RESULTS

This section presents the empirical analysis of the proposed method in terms of utility, unlearning effectiveness, time efficiency and some ablation studies.

### 5.1 EXPERIMENT SETUP

**Datasets & Models.** We conduct experiments on seven datasets: MNIST (Lecun et al., 1998), CIFAR10, CIFAR100 (Krizhevsky et al., 2009), ModelNet (Wu et al., 2015), Brain Tumor MRI (Wang et al., 2024), COVID-19 Radiography (Rahman, 2022) and Yahoo Answers dataset (Fu et al., 2022a). We adopt ResNet18 on the datasets MNIST, CIFAR10, CIFAR100, ModelNet, Brain Tumor MRI and COVID-19 Radiography. We adopt MixText (Chen et al., 2020) on the Yahoo Answers dataset. Additionally, we extend our experiments with Vgg16 on the dataset CIFAR10 and CIFAR100. Experiments are repeated over five random trials, and results are reported as mean and standard deviation. We conduct experiments on a single NVIDIA A100 GPU.

**VFL Setting & Unlearning Scenarios.** We simulate a VFL setting with one active model owner and 1–8 passive model owners. In *single-label unlearning*, one label is removed from all datasets; in *two-label unlearning*, two labels are removed from CIFAR10/100; and in *multi-label unlearning*, four labels are removed from CIFAR100. Appendix A.8 summarizes model names, VFL configurations, datasets, unlearned labels and hyper-parameters used for unlearning.

**Evaluations Metrics.** We evaluate the utility of unlearning by measuring the accuracy of remaining data, $\mathcal{D}^r$, before and after unlearning. The higher accuracy on $\mathcal{D}^r$ indicates stronger utility. A stronger utility indicates that more information about $\mathcal{D}^r$ is being preserved.

To assess unlearning effectiveness, we use a basic Membership Inference Attack (MIA) from (Shokri et al., 2017) to measure the Attack Success Rate (ASR) and the prediction accuracy on the unlearned label $y^u$ before and after unlearning. MIA determines whether a data point was part of the model's training. Lower accuracy on $y^u$ suggests more effective unlearning. A 0% ASR may indicate the Streisand effect (Golatkar et al., 2020a), where the model consistently mispredicts all $y^u$ samples as a single incorrect label. Ideally, the ASR should be slightly lower than that of a retrained model, signalling successful unlearning without revealing extra information.

Time efficiency is measured by each method's runtime, where the shorter the better. An effective unlearning method should: a) preserve $\mathcal{D}^r$ accuracy; b) reduce $y^u$ accuracy to near 0%; c) achieve an ASR slightly below that of a retrained model; and d) run quickly.

**Baselines.** We compare our method with the following baselines: Retrain, Fine-Tuning (Golatkar et al., 2020a; Jia et al., 2023), Fisher Forgetting (Golatkar et al., 2020a), Amnesiac Unlearning (Graves et al., 2021), UNSIR (Tarun et al., 2024), Boundary Unlearning (Chen et al., 2023) and SSD (Foster et al., 2024b). The implementation details of each baseline are in Appendix A.8.

### 5.2 EXPERIMENTAL RESULTS

#### 5.2.1 UTILITY GUARANTEE

To assess the utility of our proposed method, we evaluate accuracy on $\mathcal{D}^r$ before and after unlearning (Table 1). An effective unlearning method should retain as much information as possible from $\mathcal{D}^r$.

From Table 1, we observe: i) Fine-tuning preserves $\mathcal{D}^r$ well but has low unlearning effectiveness (see Section 5.2.2); ii) Fisher forgetting severely degrades $\mathcal{D}^r$ accuracy; iii) Amnesiac's random mislabeling shifts decision boundaries, hurting $\mathcal{D}^r$, especially in label-rich datasets like CIFAR100 and ModelNet; iv) UNSIR's repair step fails to fully retain $\mathcal{D}^r$, causing some degradation; v) Boundary unlearning yields inconsistent results across datasets and models; vi) While SSD preserves $\mathcal{D}^r$ reasonably well, it underperforms slightly compared to our approach; vii) In contrast, our method consistently achieves strong unlearning and retention performance.

#### 5.2.2 UNLEARNING EFFECTIVENESS

To assess unlearning effectiveness, we run MIA to determine whether the unlearned model leaks information about $y^u$ by measuring the accuracy of $y^u$ before and after unlearning.

From Tab. 1, we observe: i) Fine-tuning performs poorly on CIFAR10/100; ii) Fisher forgetting, Amnesiac, and UNSIR effectively reduce $y^u$ accuracy to 0.00%; iii) Boundary unlearning is inconsistent across datasets and models, with mixed results; iv) SSD shows strong unlearning effectiveness across all settings; v) In contrast, our method consistently achieves effective unlearning across all settings.

From Table 1, we further observe: i) Fine-tuning yields consistent ASR; ii) Fisher forgetting often shows high ASR; iii) Amnesiac consistently has low ASR; iv) UNSIR shows high ASR in most cases; v) Boundary unlearning has relatively stable ASR; vi) SSD achieves strong ASR performance across most datasets, with the exception of ModelNet and Brain MRI; and vii) our method consistently achieves strong ASR across all settings.

#### 5.2.3 TIME EFFICIENCY

Fig. 2 shows single-label unlearning runtime on CIFAR10 with ResNet18 across varying numbers of passive parties: i) Methods using the full dataset or $\mathcal{D}^r$ (*e.g.*, FT, Amnesiac, Fisher Forgetting) have high execution time; ii) Methods using only $\mathcal{D}_u$, like Boundary Unlearning, are faster; iii) Our solution has the lowest runtime (16x - 1200x lower).

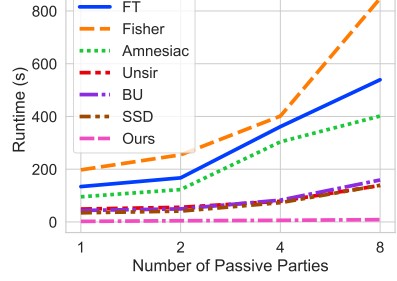

Also, the manifold mixup step is executed on the embeddings of the same passive party (see Figure 1 and Algorithm 1). As a result, the unlearning runtime increases linearly with the number of passive parties. The unlearning runtimes of different methods are compared for varying numbers of passive parties in Figure 2, demonstrating that our method remains the most efficient compared to the alternatives.

Figure 2: The runtime(s) of each unlearning method in seconds.

#### 5.2.4 OTHER MODALITY

Beyond image datasets, our method also maintains strong performance on text datasets. This demonstrates its robustness across modalities, which is particularly relevant since VFL is widely adopted in e-commerce scenarios where text is the primary modality.

Table 2 shows the effectiveness of our method in preserving the accuracy of the retained data ($\mathcal{D}^r$) while substantially reducing the accuracy of the unlearned data ($y^u$). For example, the accuracy of the unlearned data drops sharply from 41.63% to 1.41%, whereas the accuracy of the retained data decreases by less than 2%. Other baseline methods are excluded from direct comparison

| Metrics | Accuracy (%) | | |
|---|---|---|---|
| | Baseline | Retrain | Ours |
| $\mathcal{D}^r$ | 62.92 | $63.14 \pm 0.45$ | $61.01 \pm 0.91$ |
| $y^u$ | 41.63 | $0.00 \pm 0.00$ | $1.41 \pm 0.35$ |

Table 2: Single-label unlearning scenario on Yahoo Answer dataset with MixText architecture.

as they are designed primarily for image datasets and do not generalize well to text-based tasks. In conclusion, our method consistently lowers leakage and maintains stable unlearning effectiveness across diverse domains.

## 5.3 ABLATION STUDY

In this section, we conduct an ablation study on the effectiveness of different sizes of $\mathcal{D}^{p,u}$, our method across varying numbers of passive parties and different privacy-preserving VFL mechanisms.

**Module-Omission Experiments**: We conduct module-omission experiments to quantify the contribution of individual components to overall performance. We evaluate four experimental variants: (i) standard gradient ascent using all data samples, (ii) few-shot gradient ascent under the same data samples as our method, (iii) few-shot gradient ascent with mixup, and (iv) our proposed approach.

This set of experiments used ResNet18 architecture on the CIFAR-10 dataset in a single-label unlearning scenario. In Figure 3, GA-A (Gradient Ascent-All) refers to the use of all 5000 samples from $y^u$ to perform gradient ascent. GA-S (Gradient Ascent-Small) uses only 40 data samples from $y^u$, matching our method's setting, but without the Manifold Mixup and Remained

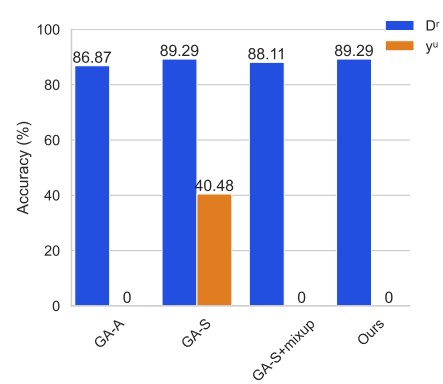

Figure 3: Comparison of the utility and unlearning effectiveness on different sizes of $\mathcal{D}^{p,u}$.

Accuracy Recovery modules. GA-S + Manifold Mixup, which corresponds to our method without the Remained Accuracy Recovery component. "Ours" refers to our full method, which also uses 40 data samples but incorporates all modules.

As shown in the Figure 3:

i GA-A effectively unlearns $y^u$ but significantly degrades the performance on $\mathcal{D}^r$.

ii GA-S preserves $\mathcal{D}^r$ well but fails to effectively unlearn $y^u$.

iii GA-S + Manifold Mixup demonstrates improved unlearning performance but still results in moderate degradation on $\mathcal{D}^r$.

iv Our full method achieves both successful unlearning of $y^u$ and strong performance preservation on $\mathcal{D}^r$, validating the contribution of the Remained Accuracy Recovery module.

**Evaluation for different numbers of passive parties**: The robustness of our approach is unaffected by the number of passive parties. Experiments with varying participant counts (see Figure 4) show consistent performance across all settings, demonstrating that scalability is preserved.

**Evaluation for different privacy-preserving VFL methods**: Our method remains effective when applied to privacy-preserving VFL models. To validate this, we evaluate performance under Differential Privacy and Gradient Compression, two widely adopted mechanisms in VFL training (Fu et al., 2022b). The results confirm that our approach maintains robustness, underscoring its practicality for real-world deployments. Please refer to Appendix A.5 for a complete result.

**Multi-label Unlearning**: Similarly, our method remains highly effective and maintains strong utility even under more challenging conditions (*i.e.* unlearning multiple labels), underscoring its reliability

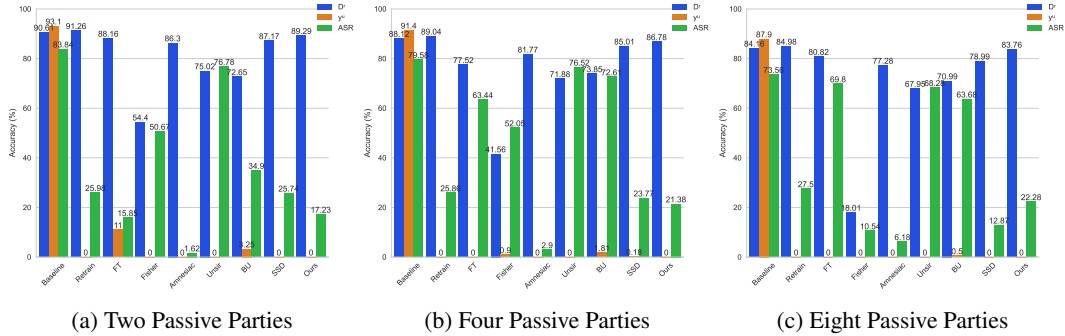

|  (a) Two Passive Parties | (b) Four Passive Parties | (c) Eight Passive Parties |

Figure 4: Accuracy of $\mathcal{D}^r$, $y^u$ and ASR for each unlearning method across ResNet18 model in single-label unlearning on different numbers of passive parties.

| Model | Datasets | Metrics | Accuracy (%) | | | | | | | | |
|---|---|---|---|---|---|---|---|---|---|---|---|
| | | | Baseline | Retrain | FT | Fisher | Amnesiac | Unsir | BU | SSD | Ours |
| ResNet18 | CIFAR10 | $\mathcal{D}^r$ | 91.48 | 91.74 ± 0.01 | **90.63 ± 0.57** | 31.25 ± 2.23 | 86.16 ± 0.82 | 74.48 ± 0.06 | 81.64 ± 0.56 | 88.17 ± 0.81 | 88.75 ± 0.36 |
| | | $y^u$ | 88.40 | 0.00 ± 0.00 | 41.15 ± 1.55 | 49.55 ± 0.40 | **0.00 ± 0.00** | **0.00 ± 0.00** | 19.90 ± 0.85 | 4.55 ± 0.31 | 0.43 ± 0.11 |
| | | ASR | 79.61 | 21.66 ± 0.64 | **13.22 ± 0.37** | 25.60 ± 0.08 | **1.84 ± 0.13** | 41.79 ± 1.35 | 35.40 ± 1.54 | 35.60 ± 0.17 | 24.21 ± 1.04 |
| | CIFAR100 | $\mathcal{D}^r$ | 71.56 | 71.21 ± 0.13 | 66.04 ± 0.58 | 53.56 ± 2.54 | 59.52 ± 0.03 | 58.02 ± 0.37 | 56.37 ± 0.39 | 66.12 ± 2.98 | **67.93 ± 0.26** |
| | | $y^u$ | 71.00 | 0.00 ± 0.00 | 38.00 ± 0.01 | 25.20 ± 5.75 | **0.00 ± 0.00** | **0.00 ± 0.00** | 13.00 ± 0.01 | **0.00 ± 0.00** | 1.17 ± 0.29 |
| | | ASR | 88.60 | 21.60 ± 0.85 | 19.20 ± 1.20 | 48.90 ± 0.54 | **6.50 ± 0.40** | 54.83 ± 0.44 | 13.70 ± 0.90 | 8.03 ± 0.54 | **6.10 ± 0.66** |
| Vgg16 | CIFAR10 | $\mathcal{D}^r$ | 89.80 | 91.13 ± 0.03 | 88.09 ± 0.35 | 47.53 ± 2.38 | 86.16 ± 0.19 | 71.50 ± 0.07 | 88.67 ± 0.22 | 86.99 ± 0.99 | **88.82 ± 0.39** |
| | | $y^u$ | 89.10 | 0.00 ± 0.00 | 28.55 ± 0.33 | 13.10 ± 0.28 | **0.00 ± 0.00** | **0.00 ± 0.00** | 19.08 ± 0.53 | 3.25 ± 1.09 | **0.00 ± 0.00** |
| | | ASR | 82.64 | 28.31 ± 1.23 | **17.75 ± 2.22** | 68.43 ± 1.14 | 46.21 ± 0.72 | 46.21 ± 0.72 | 11.72 ± 0.07 | **5.27 ± 0.01** | 28.27 ± 1.51 |
| | CIFAR100 | $\mathcal{D}^r$ | 65.75 | 65.59 ± 0.17 | 60.79 ± 0.37 | 35.24 ± 2.21 | 57.86 ± 0.81 | 56.04 ± 0.44 | 50.02 ± 0.18 | 58.97 ± 0.05 | **63.40 ± 0.13** |
| | | $y^u$ | 58.50 | 0.00 ± 0.00 | 11.75 ± 1.25 | 11.00 ± 4.85 | **0.00 ± 0.00** | **0.00 ± 0.00** | 3.25 ± 0.25 | **0.00 ± 0.00** | **0.00 ± 0.00** |
| | | ASR | 73.60 | 30.55 ± 0.05 | **22.75 ± 1.05** | 32.60 ± 1.17 | **3.45 ± 0.65** | 52.40 ± 0.80 | 27.90 ± 1.20 | 8.30 ± 0.09 | 30.47 ± 3.11 |

Table 3: Accuracy of $\mathcal{D}^r$, $y^u$ and ASR for each unlearning method across ResNet18 and Vgg16 models in two-labels unlearning

across diverse scenarios. Tables 3 and 4 present the experimental results for the two-label and multi-label unlearning scenarios, respectively. Please refer to Appendix A.6 for a detailed explanation.

**Ablation Study for** $\lambda$: For each dataset, we augment the embeddings with two coefficients, *i.e.*, $\lambda = 0.25$ and $\lambda = 0.5$. Additionally, we evaluate the impact of different $\lambda$ values in Table 5. The results indicate that variations in $\lambda$ have a minimal impact on the effectiveness of unlearning.

In summary, these ablation results demonstrate that our method remains stable across various module configurations, numbers of passive parties, privacy-preserving VFL mechanisms, multi-label unlearning, and a

| $\lambda$ Rate | Metrics | Accuracy (%) |
|---|---|---|
| [0.2, 0.5] | $\mathcal{D}_r$ | 89.11 ± 0.11 |
| | $y^u$ | 0.00 ± 0.00 |
| [0.25, 0.5] | $\mathcal{D}_r$ | 89.29 ± 0.19 |
| | $y^u$ | 0.00 ± 0.00 |
| [0.33, 0.5] | $\mathcal{D}_r$ | 88.91 ± 0.21 |
| | $y^u$ | 0.21 ± 0.02 |

Table 5: Different lambda rates on single-label unlearning scenarios on CIFAR10 dataset with ResNet18 architecture. We unlearn label "0" in this experiment.

wide range of mixup coefficients. Beyond these behavioural and robustness analyses, an important complementary question is *how much information the unlearning procedure itself reveals to passive parties ?* We therefore next examine transcript-level leakage through the lens of process privacy.

## 5.4 PROCESS PRIVACY IN VERTICAL FL UNLEARNING

Unlearning data from vertically partitioned passive models introduces a distinct privacy challenge in VFL. That is, during an unlearning request, the active party must exchange intermediate information, such as embeddings or gradients, that can inadvertently reveal which samples or labels are being deleted. For example, in retraining, the active party must indicate the

| Model | Datasets | Metrics | Accuracy (%) | | | | | | | | |
|---|---|---|---|---|---|---|---|---|---|---|---|
| | | | Baseline | Retrain | FT | Fisher | Amnesiac | Unsir | BU | SSD | Ours |
| ResNet18 | CIFAR100 | $\mathcal{D}^r$ | 71.53 | $71.91 \pm 0.12$ | $67.16 \pm 0.13$ | $54.79 \pm 1.04$ | $59.09 \pm 0.54$ | $59.05 \pm 0.38$ | $48.96 \pm 0.04$ | $67.09 \pm 1.11$ | $\mathbf{69.97 \pm 0.03}$ |
| | | $y^u$ | 72.00 | $0.00 \pm 0.00$ | $33.87 \pm 0.88$ | $45.38 \pm 1.13$ | $\mathbf{0.00 \pm 0.00}$ | $\mathbf{0.00 \pm 0.00}$ | $15.00 \pm 0.25$ | $0.11 \pm 0.07$ | $0.33 \pm 0.14$ |
| | | ASR | 86.65 | $16.95 \pm 0.35$ | $18.23 \pm 1.63$ | $62.78 \pm 3.93$ | $\mathbf{6.05 \pm 1.19}$ | $68.63 \pm 1.83$ | $38.35 \pm 0.75$ | $\mathbf{3.33 \pm 0.54}$ | $12.40 \pm 1.12$ |
| Vgg16 | CIFAR100 | $\mathcal{D}^r$ | 65.83 | $65.66 \pm 0.08$ | $60.92 \pm 0.08$ | $36.55 \pm 1.07$ | $57.26 \pm 0.18$ | $56.86 \pm 0.26$ | $47.04 \pm 0.32$ | $61.01 \pm 0.55$ | $\mathbf{64.44 \pm 0.12}$ |
| | | $y^u$ | 60.25 | $0.00 \pm 0.00$ | $7.63 \pm 0.13$ | $28.75 \pm 1.25$ | $\mathbf{0.00 \pm 0.00}$ | $\mathbf{0.00 \pm 0.00}$ | $7.13 \pm 0.11$ | $\mathbf{0.00 \pm 0.00}$ | $1.17 \pm 0.25$ |
| | | ASR | 75.80 | $27.20 \pm 0.75$ | $\mathbf{24.38 \pm 3.13}$ | $55.20 \pm 3.75$ | $\mathbf{4.80 \pm 0.05}$ | $32.83 \pm 0.58$ | $29.70 \pm 0.03$ | $10.98 \pm 0.59$ | $27.93 \pm 0.29$ |

Table 4: Accuracy of $\mathcal{D}^r$, $y^u$ and ASR for each unlearning method across ResNet18 and Vgg16 models in multi-labels unlearning

sample IDs to be removed, directly exposing the deletion set. In boundary unlearning, gradients associated with the deleted label may similarly disclose sensitive membership information. As shown in Table 6, retraining reveals the full deletion set to the passive party (100% leakage), and the risk increases further when multiple labels are unlearned (Appendix A.3).

To make this leakage channel explicit, we introduce the notion of *process privacy*, which characterises how much the passive party's belief about the deletion set can change after observing the protocol transcript. Let the passive party begin with a prior belief about which samples constitute the deletion set. After receiving the transcript $\mathcal{T}$, which may contain embeddings, gradients, or auxiliary updates, the passive party updates this belief. We quantify this shift using the Kullback–Leibler divergence between the posterior and prior beliefs. The smaller the divergence, the less information the transcript conveys about the deleted data.

| Datasets | Membership Leakage Rate (%) | |
|---|---|---|
| | Standard Retraining | Ours |
| CIFAR10 | 100 | 14.38 |
| CIFAR100 | 100 | 4.04 |

Table 6: The averaged membership leakage rate evaluated across all class for CIFAR10/100 with ResNet18 (lower better). We set $k = 5000$ for CIFAR10 and $k = 500$ for CIFAR100, where $k$ is the total number of samples in each class.

**Definition 1** (**Process Privacy**). *A VFU protocol satisfies process privacy with parameter $\varepsilon$ if the passive party's posterior belief about the deletion set, after observing the transcript $\mathcal{T}$, differs from its prior belief by at most $\varepsilon$, measured via the Kullback–Leibler divergence. Formally, the protocol satisfies process privacy when $D_{\mathrm{KL}}(P(\mathcal{D}^u \mid \mathcal{T}) \parallel P(\mathcal{D}^u)) \leq \varepsilon$, which captures the requirement that the unlearning transcript should not substantially increase the passive party's ability to infer which samples were deleted.*

This formulation aligns naturally with our empirical leakage evaluation, which quantifies how the passive party's inference capability changes after observing the transcript under retraining, boundary unlearning, and our method. Our approach restricts information exchange to embeddings and gradient updates of a small public subset, which, supported by the Bernstein bound in Appendix A.4 limits the residual influence of unlearned data. Empirically, this yields a substantial reduction in leakage, from 100% under retraining to as low as 14.38% under our method, as shown in Table 6.

As VFU is an emerging research area, prior work has not formally characterised this transcript-level privacy dimension. To the best of our knowledge, this paper is among the first to articulate, define, and empirically study process privacy in the VFL context. While the definition provides a conceptual foundation for describing transcript leakage, deriving tight theoretical upper bounds on $\varepsilon$ remains significantly more challenging and is left as an open problem.

## 6 CONCLUSION

This paper presents a pioneering approach to label unlearning within the vertical federated learning (VFL) domain, addressing a critical gap in the existing literature. By introducing a few-shot unlearning method that leverages manifold mixup, we effectively mitigate the risk of label privacy leakage while ensuring efficient unlearning from both active and passive models. Our systematic exploration of potential label privacy risks and extensive experimental validation on benchmark datasets underscore the proposed method's efficacy and rapid performance. Ultimately, this work not only advances the understanding of unlearning in VFL but also sets the stage for further innovations in privacy-preserving collaborative machine learning practices.

ACKNOWLEDGEMENT

This research is supported by the ASEAN-China Cooperation Fund (ACCF) under the project "*Deep Ensemble Under Non-Ideal Conditions and Its Typical Applications in Computer Vision*".

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

## A    APPENDIX

This appendix provides additional theoretical insights, extended experimental results and further implementation details to support the main submission. We begin with a theoretical analysis of unlearning effectiveness in Section A.1, followed by a table of notations in Section A.2 for reference. Section A.3 provides an explanation of label leakage during the unlearning process in VFL, while Section A.4 discusses why our method can achieve strong results with only a small number of data samples. In addition, Section A.5 presents ablation experiments on applying our method to privacy-preserved VFL models. Section A.6 presents the experimental results for two-label and multi-label unlearning scenarios, showcasing the effectiveness of our solution in handling more complex unlearning tasks. This section also includes a detailed performance analysis of each baseline across three key metrics: (i) accuracy on $\mathcal{D}^r$, (ii) accuracy on $y^u$ and (iii) ASR score. Section A.7 presents a comparative table of our method against existing Federated Unlearning approaches, highlighting the capability of our method to preserve label privacy during the unlearning process. In Section A.8, we provide detailed information about the experiment setup, including model architectures, unlearning scenarios, baseline methods, datasets, and the data distribution among parties. Finally, in Section A.9, we discuss the limitations of our work, including aspects beyond its scope, methodological constraints, and other relevant considerations.

### A.1    THEORETICAL ANALYSIS FOR UNLEARNING EFFECTIVENESS

**Theorem 2.** *Suppose that the trained passive model $\theta$ and active model $\omega$ achieve near-zero training loss. Then, when unlearning a single label, the following holds:*

$$
\begin{aligned}
\mathbb{E}_{(\vec{H}^u, \vec{y}^u)} \nabla_\omega \ell(\omega; \vec{H}^u, \vec{y}^u) \cdot \mathbb{E}_{(H^u, y^u)} \nabla_\omega \ell(\omega; H^u, y^u) > 0, \\
\mathbb{E}_{(\vec{H}^u, \vec{y}^u)} \nabla_\theta \ell(\theta; \vec{H}^u, \vec{y}^u) \cdot \mathbb{E}_{(H^u, y^u)} \nabla_\theta \ell(\theta; H^u, y^u) > 0,
\end{aligned}
\tag{8}
$$

*where $(\vec{H}^u, \vec{y}^u)$ denotes the manifold mixup embeddings and labels of the public data $\mathcal{D}^{p,u}$ associated with the unlearned label, $(H^u, y^u)$ denotes the embeddings and labels of the complete unlearned dataset $\mathcal{D}^u$., and $\ell$ is the main task loss of VFL.*

*Proof.* In simplified analysis, we set the loss based on the small-size public data $\mathcal{D}^{u,p}$, the manifold mixup of $\mathcal{D}^{u,p}$, and the original unlearned data $\mathcal{D}^u$ are $\ell_1$, $\ell_{mix}$ and $\ell_2$ respectively. We consider a two-layer linear neural network: $f(x) = \omega(\theta x)$. The loss function is the Mean Squared Error (MSE): $\ell = \frac{1}{2|D|} \sum_{i \in D} \|f(x_i) - y_i\|^2$.

For a public unlearned sample $(x_i^u, y_i^u)$, let $H_i^u = \theta x_i^u$ be the hidden representation, $z_i = \omega H_i^u$ be the output, $e_i = z_i - y_i^u$ be the error. The gradients are:

$$
\nabla_\omega \ell_i = e_i H_i^{u\top}, \quad \nabla_\theta \ell_i = \omega^\top e_i x_i^{u\top}.
\tag{9}
$$

Averaging the gradients over the any dataset $\mathcal{D}^{u,p}$ are:

$$
\nabla_\omega \ell = \frac{1}{|D|} \sum_i e_i H_i^{u\top}, \quad \nabla_\theta \ell = \frac{1}{|D|} \sum_i \omega^\top e_i x_i^{u\top}
\tag{10}
$$

Given two samples $(x_i^u, y_i^u)$, $(x_j^u, y_j^u)$, and a mixup coefficient $\lambda \in [0, 1]$, we have:

$$
\begin{aligned}
H_{\text{mix}} = \lambda H_i^u + (1-\lambda) H_j^u, \quad y_{\text{mix}} = \lambda y_i^u + (1-\lambda) y_j^u, \\
z_{\text{mix}} = \omega H_{\text{mix}} = \lambda z_i + (1-\lambda) z_j^u e_{\text{mix}} = z_{\text{mix}} - y_{\text{mix}} = \lambda e_i + (1-\lambda) e_j
\end{aligned}
\tag{11}
$$

Then we have the gradients on $\omega$ for manifold mixup:

$$
\begin{aligned}
\nabla_\omega \ell_{\text{mix}} &= e_{\text{mix}} H_{\text{mix}}^\top \\
&= (\lambda e_i + (1-\lambda) e_j)(\lambda H_i^u + (1-\lambda) H_j^{u\top}) \\
&= \lambda^2 e_i H_i^{u\top} + \lambda(1-\lambda)(e_i H_j^{u\top} + e_j H_i^{u\top}) + (1-\lambda)^2 e_j H_j^{u\top}
\end{aligned}
$$

The gradients on $\theta$ for manifold mixup:

$$
\begin{aligned}
\nabla_\theta \ell_{\text{mix}} &= \nabla_h \ell_{\text{mix}} \cdot \nabla_\theta H_{\text{mix}} \\
&= \omega^\top e_{\text{mix}} (\lambda x_i^{u\top} + (1-\lambda) x_j^{u\top}) \\
&= \lambda^2 \omega^\top e_i x_i^{u\top} + \lambda(1-\lambda)(\omega^\top e_i x_j^{u\top} + \omega^\top e_j x_i^{u\top}) + (1-\lambda)^2 \omega^\top e_j x_j^{u\top}
\end{aligned}
$$

Since

$$
\mathbb{E}[eh^\top] = \nabla_\omega \ell_1, \quad \mathbb{E}[\omega^\top e x^\top] = \nabla_\theta \ell_1
$$

We have:

$$
\mathbb{E}[\nabla_\omega \ell_{\text{mix}}] = 2\mathbb{E}[\lambda^2]\nabla_\omega \ell_1 + 2(\mathbb{E}[\lambda] - \mathbb{E}[\lambda^2])\mathbb{E}[e]\mathbb{E}[h]^\top
$$

$$
\mathbb{E}[\nabla_\theta \ell_{\text{mix}}] = 2\mathbb{E}[\lambda^2]\nabla_\theta \ell_1 + 2(\mathbb{E}[\lambda] - \mathbb{E}[\lambda^2])\omega^\top \mathbb{E}[e]\mathbb{E}[x]^\top
$$

Due to the near-zero training loss, $\mathbb{E}[e]$ tends to be zero, we obtain the averaged gradients of $\omega$ and $\theta$ on the public dataset $\mathcal{D}^{u,p}$

$$
\mathbb{E}[\nabla_\omega \ell_{\text{mix}}] \sim 2\mathbb{E}[\lambda^2]\nabla_\omega \ell_1, \quad \mathbb{E}[\nabla_\theta \ell_{\text{mix}}] \sim 2\mathbb{E}[\lambda^2]\nabla_\theta \ell_1 \tag{12}
$$

Since the $\mathcal{D}^{p,u}$ is the subset of the total unlearned dataset $\mathcal{D}^u$, we can leverage the Chebyshev's Inequality to obtain $\nabla_\theta \ell_1 \cdot \nabla_\theta \ell_2 > 0$ and $\nabla_\omega \ell_1 \cdot \nabla_\omega \ell_2 > 0$ with the probability $1 - O(\frac{1}{n_{p,u}})$, where $n_{p,u}$ is the data size of $\mathcal{D}^{p,u}$. Combining Eq.equation 12, we can obtain

$$
\mathbb{E}[\nabla_\omega \ell_{\text{mix}}] \cdot \mathbb{E}[\lambda^2]\nabla_\omega \ell_2 > 0, \quad \mathbb{E}[\nabla_\theta \ell_{\text{mix}}] \cdot \mathbb{E}[\lambda^2]\nabla_\theta \ell_2 > 0, \tag{13}
$$

which completes the proof. $\qquad\square$

Theorem 2 indicates that the gradient update direction for label unlearning, when applied to the augmented embeddings of the public unlearned data, is positively aligned with the update direction derived from the embeddings of the entire unlearned dataset. This result suggests that gradient-based label unlearning using only public data is effective and approximates the behavior of unlearning with access to all unlearned data.

## A.2 TABLE OF NOTATION

Table 7 summarises all notations used throughout the paper for clarity.

| Notation | Meaning |
|---|---|
| $F_\omega, G_{\theta_k}$ | Active model and $k_{th}$ passive model |
| $w^u, \theta_k^u$ | Unlearned active model and unlearned $k_{th}$ passive model |
| $K$ | The number of passive party |
| $\lambda$ | Mixed coefficient |
| $\eta$ | Learning rate |
| $N$ | Unlearning epochs |
| $\mathbf{x}_k$ | Private features own by $k_{th}$ passive party |
| $y$ | Private label owned by active party |
| $y^u$ | The unlearn labels |
| $\{x_k^u\}$ | The unlearned feature for client $k$ corresponding to the $y^u$ |
| $x_k^p$ | The known features for client $k$ corresponding to the $y^u$ |
| $H_k$ | Forward embedding of passive party $k$ |
| $\vec{H}_k^u, \vec{H}_k^r$ | Augmented forward embedding of passive party $k$ of unlearn data and remain data. |

Table 7: Table of Notations

## A.3 LABEL LEAKAGE DURING UNLEARNING

Figure 5 shows the label leakage (in %) of Boundary Unlearning in VFL settings for varying numbers of unlearning labels. Single label unlearning shows 100% label leakage since passive parties can

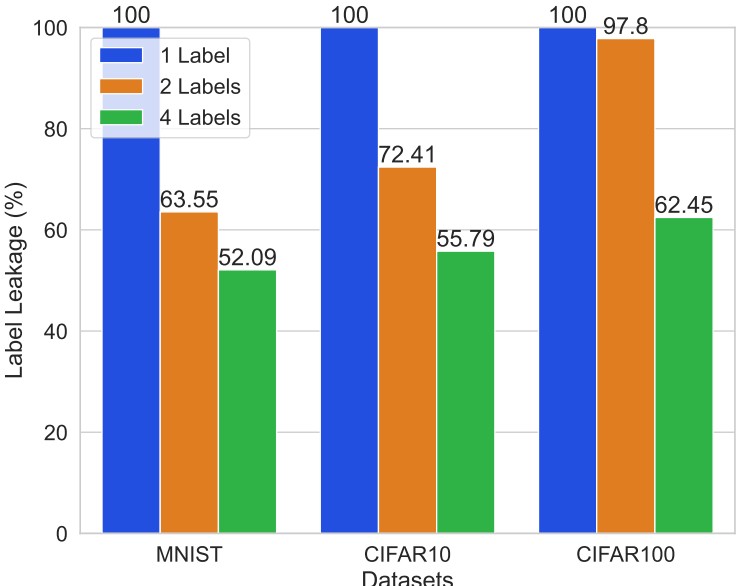

Figure 5: Illustration of label leakage (%) with Boundary unlearning in VFL using ResNet18 model on different numbers of labels and datasets.

immediately infer that all samples involved belong to the same label. Moreover, when unlearning multiple labels ($m_u$), label leakage worsens as the passive party can infer on the transferred gradient by the active party. Increasing the number of labels, therefore, results in a larger portion of data being exposed during the unlearning process. For instance, with four labels from CIFAR-100, a total of 62.45% of label leakage is exposed. In boundary unlearning, the passive party can infer label information from the gradients $\mathbf{g}^u$ sent by the active party. Specifically, the passive party employs clustering on $\mathbf{g}^u$ to derive $m_u$ clusters by optimizing the following objective function:

$$\min \sum_{g_i \in \mathcal{C}_j} \sum_{j=1}^{m_u} |g_i^u - \bar{g}_j^u|, \tag{14}$$

where $\mathcal{C}_j$ denotes the set of points assigned to cluster $j$, and $\bar{g}_j^u$ represents the centroid of cluster $j$. Consequently, the passive party can deduce the labels of the features in $\mathcal{X}$.

## A.4 WHY TINY PUBLIC SET SUFFICES

Our goal is *unlearning*, not full re-training. Thus, we only need a small parameter shift to cancel the target label's influence. Below, we quantify three reasons why a much smaller public set $D^{u,p}$ can achieve this.

1. **Variance reduction via manifold mixup.** Manifold mixup augments each public example into an infinite family of virtual points, reducing the gradient-estimator variance proportionally to the number of synthetic mixtures. Recent few-shot studies show that manifold-mixup greatly reduces sample requirements. For example, (Mangla et al., 2020) achieves higher accuracy than prior baselines across four vision benchmarks with only 5 to 20 samples per class. Similarly, SimpliMix (Yang et al., 2024) achieves strong gains in few-shot 3D point cloud classification.

2. **Tiny update magnitude.** The pre-unlearning model already fits the data ($loss \approx 0$). As shown by (Chen et al., 2019), when the source and target domains are similar, even simple few-shot methods can perform well with minimal fine-tuning. Hence, once the gradient direction is reliable, only a small update is needed. In other words, because unlearning starts from a model that already fits the data, only the label-specific contribution needs to be removed, resulting in a small parameter displacement. In the quadratic loss setting analysed

in our theoretical analysis (Appendix A.1), the required step length scales with the gradient norm. Once the direction is correct, even a single mini-batch update can eliminate the label.

3. **Exponential concentration of the gradient direction.** Extending Eq. equation 12 in our Appendix A.1 with the vector Bernstein inequality (see Section 2.8 of (Vershynin, 2018)) gives

$$\Pr\!\left[\cos\angle(\nabla_\omega \ell_{\mathrm{mix},i}, \nabla_\omega \ell) \leq 1 - \varepsilon\right] \;\leq\; \exp\!\left(-c\,|D^{u,p}|\,\varepsilon^2\right),$$

where $c$ depends on $\mathbb{E}[\lambda^2]$. Thus, the chance that the mixed-sample gradient deviates substantially from the full-data gradient vanishes exponentially with the number of public samples; a few dozen already give high-probability alignment.

Specifically, the underlying tail bound for the sum of augmented embeddings follows precisely the Bernstein inequality (Vershynin, 2018):

$$P\{|\textstyle\sum_i X_i| \geq t\} \;\leq\; 2\exp[-c\min\{t^2/\textstyle\sum_i \|X_i\|_{\psi_1}^2,\, t/\max_i \|X_i\|_{\psi_1}\}],$$

where $X_i = \cos\angle(\nabla_\omega \ell_{\mathrm{mix},i}, \nabla_\omega \ell)$ and $\ell_{\mathrm{mix},i}$ represents $i_{th}$ mixture gradients.

*Intuition.* Each augmented sample acts like an arrow pointing toward the true unlearning direction. Applying manifold mixup to only a small number of data samples generates thousands of synthetic gradient directions. Averaging these directions cancels out random variance, as guaranteed by the Bernstein inequality. Because our model already starts near the solution, a short step along this accurately aligned direction suffices to complete unlearning.

## A.5   EVALUATION FOR DIFFERENT PRIVACY PRESERVING VFL METHODS

We evaluate our unlearning method under two privacy preserving VFL methods: (i) Differential Privacy (Fu et al., 2022b) and (ii) Gradient Compression (Fu et al., 2022b). Fig. 6 presents the effectiveness of our solution on both methods across different levels of variance Gaussian noise and compression ratio. Higher Gaussian noise levels and greater gradient compression ratios enhance privacy protection in VFL but lead to increased performance degradation. It shows that even for a large compression ratio and noise level, our proposed method can still unlearn effectively, while the utility of the vertical training decreases significantly.

## A.6   TWO-LABEL AND MULTI-LABEL UNLEARNING

Tables 3 and 4 present the experimental results obtained from the two-label and multi-label unlearning scenarios, respectively.

Table 3 provides a detailed breakdown of the performance metrics and outcomes observed during the two-label unlearning process across various datasets and architectures. It highlights the accuracy scores of $\mathcal{D}^r$, $y^u$ and ASR when unlearning was applied to labels *"0"* and *"2"*, demonstrating the utility guarantees and the effectiveness of the baseline unlearning methods.

Similarly, Tab. 4 summarizes the results of the multi-label unlearning experiments, where labels *"0"*, *"2"*, *"5"*, and *"7"* were unlearned for all datasets and architectures. This table captures critical metrics such as the accuracy scores of $\mathcal{D}^r$, $y^u$ and ASR reflecting the outcomes of the multi-label unlearning process.

### A.6.1   UTILITY GUARANTEE

From Tab. 3, the FT method shows performance consistent with the single-label unlearning scenario. It preserves $\mathcal{D}^r$ well on datasets with a small number of labels (*e.g.*, CIFAR-10) but performs poorly on datasets with many labels (*e.g.*, CIFAR-100). The Fisher Forgetting method consistently demonstrates poor preservation of $\mathcal{D}^r$ across all datasets and model architectures. The Amnesiac approach performs similarly to FT, achieving strong preservation of $\mathcal{D}^r$ on datasets with fewer labels (e.g., CIFAR-10) but showing reduced performance on datasets with a larger label space (e.g., CIFAR-100). The Unsir and BU methods also follow this trend, maintaining good preservation on simpler datasets like CIFAR-10, but experiencing greater degradation in $\mathcal{D}^r$ accuracy across all datasets compared to FT and Amnesiac. SSD demonstrates strong preservation of $\mathcal{D}^r$ across all experiments, except for the CIFAR100 dataset under the VGG16 architecture.

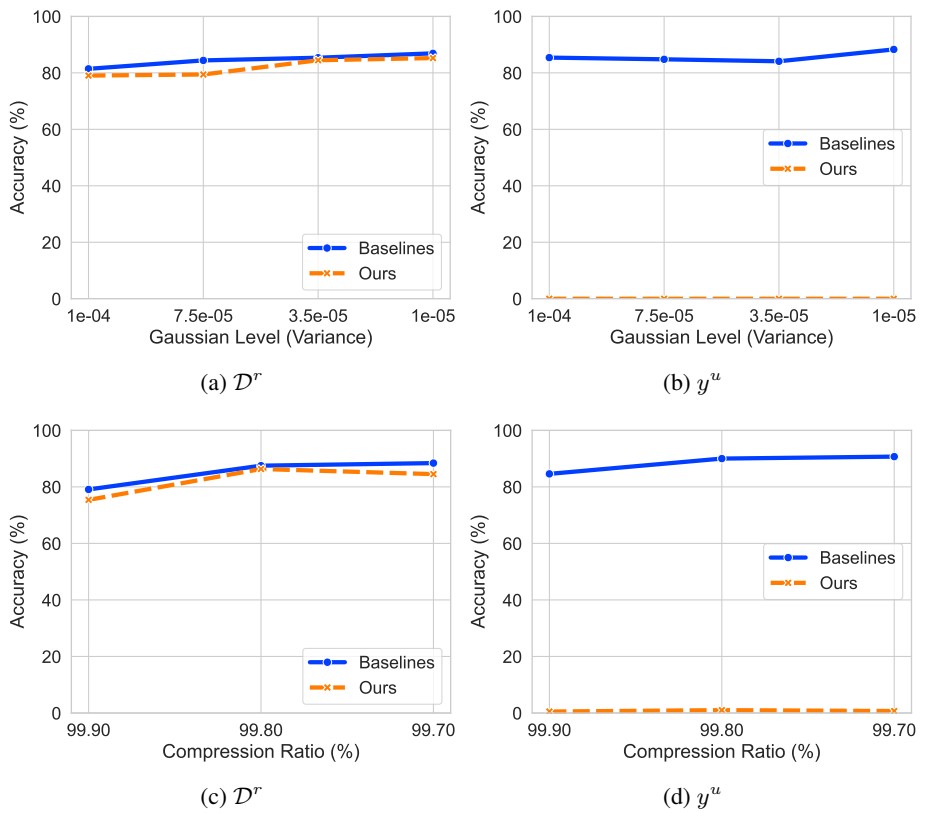

Figure 6: Comparison of remained accuracy $\mathcal{D}^r$ and unlearned accuracy $y^u$ under Gaussian noise variance (a, b) and gradient compression ratio (c, d). Subplots (a, c) represent the remained accuracy $\mathcal{D}^r$, while (b, d) show the unlearned accuracy $y^u$.

In contrast, our method consistently achieves strong unlearning utility while perfectly preserving $\mathcal{D}^r$ across all experimental settings.

From Tab. 4, the FT method shows similar behavior to the single-label and two-label unlearning scenarios, with poor preservation of $\mathcal{D}^r$ on datasets that have a large number of labels (*e.g.*, CIFAR-100). The Fisher Forgetting method consistently performs poorly in preserving $\mathcal{D}^r$ across all datasets and model architectures. The Amnesiac approach also mirrors the performance of FT, showing reduced preservation on datasets with many labels. Similarly, the Unsir and BU methods exhibit even greater degradation in $\mathcal{D}^r$ performance across all datasets compared to FT and Amnesiac. SSD yields comparable results in both single-label and two-label unlearning scenarios, showing strong preservation of $\mathcal{D}^r$, though its performance is slightly inferior to ours.

In contrast, our method demonstrates strong unlearning utility and achieves near-perfect preservation of $\mathcal{D}^r$ across all experimental settings, including multi-label unlearning scenarios and for datasets with a large number of labels.

In conclusion, our experimental results across *single-label*, *two-label*, and *multi-label* unlearning tasks demonstrate that our method consistently delivers the best balance between utility and privacy preservation. For example, unlike FT, which maintains $\mathcal{D}^r$ accuracy only on datasets with few labels (*e.g.*, CIFAR-10) but suffers on CIFAR-100, our method achieves near-perfect preservation of $\mathcal{D}^r$ regardless of label complexity. Fisher Forgetting performs poorly across all datasets and architectures, leading to significant drops in retained accuracy. Amnesiac Unlearning introduces instability through random relabeling of $\mathcal{D}_u$, leading to accuracy degradation on high-label datasets, whereas Unsir and BU show even greater declines, especially in multi-label settings. SSD shows good preservation of $\mathcal{D}^r$ across different settings, but its performance is slightly inferior to ours. In contrast, our method maintains high $\mathcal{D}^r$ accuracy (often above 98%) while reducing unlearned data accuracy close to random chance, consistently across all scenarios. These results provide strong

evidence that our approach offers superior robustness, effectiveness, and generalizability for practical, privacy-preserving unlearning.

### A.6.2 UNLEARNING EFFECTIVENESS

We evaluate unlearning effectiveness using two key metrics: the accuracy on $y^u$ and the Attack Success Rate (ASR). A lower accuracy on $y^u$ indicates more effective unlearning. However, an ASR of 0% may signal the presence of the Streisand effect (Golatkar et al., 2020a), where the model consistently misclassifies all $y^u$ samples into the same incorrect label, which indicates potentially leaking information about the unlearned label. Ideally, the ASR should be slightly lower than that of a retrained model, reflecting successful unlearning without revealing leakage. If the model consistently assigns all $y^u$ to a specific wrong label, it might expose that the model has treated $y^u$ uniquely. It suggests the model has not truly *"forgotten"* the data but developed an unusual bias against it.

**Analysis on $y^u$** : This paragraph provides a detailed analysis of each baseline method's performance with respect to the $y^u$ accuracy metric.

From Tab. 3, both FT and Fisher Forgetting methods exhibit poor unlearning effectiveness in the two-label unlearning scenario. These methods perform effectively only in the single-label unlearning context. In contrast, both the Amnesiac and Unsir methods consistently achieve perfect unlearning effectiveness in the two-label unlearning scenario. The BU method shows a decline in unlearning effectiveness compared to its performance in single-label unlearning, indicating challenges in handling scenarios with two labels. SSD demonstrates strong unlearning effectiveness on the CIFAR100 dataset but fails to reduce the accuracy of $y^u$ to 0.00% on CIFAR10 dataset. Meanwhile, our method continues to demonstrate exceptional unlearning effectiveness, maintaining its outstanding performance in the two-label unlearning scenarios, further solidifying its robustness and efficiency.

From Tab. 4, the FT and Fisher Forgetting methods exhibit similarly poor unlearning effectiveness in the multi-label unlearning scenario, comparable to their performance in the two-label unlearning context. These methods are effective only in the single-label unlearning scenario. In contrast, the Amnesiac and Unsir methods consistently achieve perfect unlearning effectiveness in the multi-label scenario, mirroring their performance in single-label and two-label contexts. The BU method performs similarly to the two-label scenario in the multi-label context, showing limited unlearning effectiveness. SSD shows good unlearning effectiveness in a multi-label scenario. Meanwhile, our method continues to demonstrate exceptional unlearning effectiveness, maintaining outstanding performance in multi-label scenarios, further reinforcing its robustness and efficiency.

In conclusion, our method demonstrates superior performance across all unlearning scenarios. Specifically, it consistently achieves near-perfect preservation of retained data $\mathcal{D}^r$, often maintaining accuracy drops below 2% while reducing the accuracy of unlearned labels to near-random levels (*e.g.*, from 41.63% to 1.41%). In contrast, FT and Fisher Forgetting fail to scale beyond simple settings: they perform adequately only in single-label scenarios but suffer substantial degradation in $\mathcal{D}^r$ accuracy when applied to two-label or multi-label tasks, especially on datasets like CIFAR-100. Amnesiac and Unsir do achieve strong unlearning effectiveness across all scenarios, but at the cost of destabilizing the decision boundaries, which leads to marked drops in $\mathcal{D}^r$ accuracy for more complex datasets. BU also shows deteriorating performance as task complexity increases. SSD performs well in unlearning, but shows minor weakness on CIFAR10 under the two-label setting. Only our method balances both objectives, that is effectively unlearning target labels while preserving the rest, across all datasets, architectures, and label complexities. These concrete results affirm our approach as the most robust, generalizable, and practically deployable solution for modern unlearning needs.

**Analysis on ASR** : This paragraph discusses how each baseline method performs in terms of ASR, highlighting their strengths and weaknesses.

From Tab. 3 and 4, FT demonstrates excellent performance, achieving ASR scores that are consistently lower but close to those of the retrain model across all datasets, with the exception of the multi-labels scenario involving the ResNet18 architecture and the CIFAR100 dataset. In that case, the ASR score is only slightly higher than that of the retrain model. In contrast, Fisher Forgetting performs poorly in both the two-labels and multi-labels scenarios, exhibiting high ASR scores across all experiments, which indicates its ineffectiveness in mitigating adversarial risks. The Amnesiac method shows a consistent pattern of very low ASR scores across all experiments, potentially suggesting vulnerabilities

to the Streisand effect, as observed in previous analyses. The Unsir method performs similarly to Fisher Forgetting, also achieving high ASR scores across all experiments, further highlighting its ineffectiveness. Boundary Unlearning continues to exhibit inconsistent performance throughout the experiments, failing to deliver reliable results. SSD continues to perform well across the experiments, consistently achieving ASR scores lower than and close to those of the retrain model. On the other hand, our method demonstrates outstanding performance across many datasets, achieving ASR scores that are consistently lower than, and close to, those of the retrain model. This highlights our method's effectiveness and reliability in addressing adversarial risks while maintaining robust unlearning outcomes.

In conclusion, the results clearly demonstrate the effectiveness and robustness of our method in mitigating adversarial risks. Across datasets, our approach consistently achieves ASR scores that are not only low but also often lower than those of the retrained models, indicating strong unlearning without compromising security. While FT and SSD perform reasonably well in most scenarios, other methods such as Fisher Forgetting, Unsir, and Boundary Unlearning exhibit limited effectiveness, with high ASR scores and inconsistent performance. Although the Amnesiac method achieves very low ASR scores, it raises concerns about potential vulnerabilities, such as the Streisand effect. In contrast, our method strikes the optimal balance between unlearning effectiveness and adversarial robustness, making it the most reliable and secure solution among the evaluated approaches.

### A.6.3    FURTHER ANALYSIS OF BASELINES

Fine-Tuning (FT): FT demonstrates adequate performance in utility preservation for datasets with a smaller number of labels (*e.g.*, CIFAR10), its effectiveness diminishes sharply as the dataset complexity increases, failing to preserve $\mathcal{D}^r$ effectively for datasets with a large number of labels (*e.g.*, CIFAR100). Additionally, its unlearning effectiveness is limited to single-label scenarios, highlighting its inability to handle more complex unlearning contexts.

Fisher Forgetting: This method struggles across all dimensions, exhibiting poor preservation of $\mathcal{D}^r$ irrespective of the dataset or architecture. It also fails to achieve unlearning effectiveness in two-label and multi-labels unlearning scenarios, with high adversarial success rates (ASR) that compromise its ability to address adversarial risks. This consistent underperformance underscores the method's lack of robustness in diverse scenarios.

Amnesiac Unlearning: While the Amnesiac method demonstrates strong unlearning effectiveness across various scenarios, its utility preservation declines significantly when applied to datasets with a large number of labels, such as CIFAR-100. This is primarily due to its approach of randomly assigning incorrect labels to $\mathcal{D}_u$, which introduces unpredictable shifts in decision boundaries and undermines reliability in more complex settings. Additionally, the method consistently yields very low ASR scores across all experiments. This implies that although low ASR scores indicate strong unlearning, they may also indicate susceptibility to the Streisand effect, as observed in prior analyses.

Unsir and BU Methods: Both methods show similar limitations, with significant degradation in utility preservation as dataset complexity increases. Although these methods achieve moderate unlearning effectiveness, they exhibit high ASR scores, indicating their vulnerability to adversarial risks. The inconsistency of their performance across scenarios highlights their lack of generalizability and robustness.

SSD: SSD demonstrates strong performance across all three evaluation metrics, namely accuracy on $\mathcal{D}^r$, accuracy on $y^u$ and ASR score, although it remains marginally worse than our method in each case. Moreover, SSD incurs a higher computational cost, resulting in longer runtime than our approach (see Fig. 2 in Sect. 5.2.3), further limiting its practical efficiency.

Overall, the experimental results highlight the key limitations of existing methods, such as FT, Fisher Forgetting, Amnesiac, Unsir, BU and SSD in striking a balance between utility preservation, unlearning effectiveness, and low ASR scores. While some methods may perform well in one aspect, they often fall short in others, especially in complex or multi-label scenarios. In contrast, our method consistently delivers strong unlearning performance, near-perfect utility preservation, and robust ASR scores across all datasets and settings. This comprehensive and consistent performance firmly establishes our approach as the most reliable, effective, and scalable solution for tackling diverse and challenging unlearning tasks.

| Method | Type of FL | Unlearning Types | Unlearning Targets | Protection for Unlearned Data |
|---|---|---|---|---|
| FedEraser (Liu et al., 2021) | Horizontal | Exact | Client | ✗ |
| FRU (Yuan et al., 2023) | | Approximate | Client | ✗ |
| FedRecovery (Zhang et al., 2023a) | | Approximate | Client | ✗ |
| VeriFI (Gao et al., 2024) | | Approximate | Client | ✗ |
| HDUS (Ye et al., 2024) | | Approximate | Client | ✗ |
| KNOT (Su & Li, 2023) | | Approximate | Client | ✗ |
| FedRecover (Cao et al., 2023a) | | Approximate | Client | ✗ |
| Knowledge Distillation (Wu et al., 2022) | | Approximate | Client | ✗ |
| Discriminative Pruning (Wang et al., 2022) | | Approximate | Class | ✗ |
| MoDe (Zhao et al., 2024) | | Approximate | Class | ✗ |
| Rapid Retraining (Liu et al., 2022) | | Exact | Sample | ✗ |
| QuickDrop (Dhasade et al., 2023) | | Approximate | Sample | ✗ |
| FedAU (Gu et al., 2024b) | | Approximate | Class, Sample & Client | ✗ |
| Ferrari (Gu et al., 2024a) | | Approximate | Feature | ✗ |
| Fast Retraining (Wang et al., 2024) | Vertical | Exact | Passive Party | ✗ |
| SecureCut (Li et al., 2024) | | Approximate | Instance & Passive Party | ✗ |
| Constraint Imposing (Deng et al., 2023) | | Approximate | Passive Party | ✗ |
| Backdoor Certification (Han et al., 2025) | | Approximate | Passive Party | ✗ |
| **Ours** | | **Approximate** | **Label** | ✓ |

Table 8: Comparison of our method with existing studies on Federated Unlearning. This table demonstrates that our method is the ***only one*** that ensures privacy protection for $\mathcal{D}_u$ during the unlearning process. To the best of our knowledge, it is also the first approach to tackle label unlearning in VFL while safeguarding the unlearned label throughout the process.

## A.7   RELATED WORKS

Table 8 compares our method with existing studies on Federated Unlearning. Most methods target *Horizontal Federated Learning (HFL)* and focus on *Client* unlearning. A smaller set of methods addresses *Vertical Federated Learning (VFL)*, which typically involves specific unlearning for passive parties. **Privacy protection for unlearned data** is generally lacking, as shown by the prevalence of ✗ across methods. Notably, it uniquely offers **protection for unlearned data**, setting it apart from previous approaches.

## A.8   EXPERIMENT SETUP

This section provides detailed information on our experimental settings. Tables 9 and 10 summarize the hyper-parameters for our unlearning method. Table 11 summarizes the model name, VFL framework settings, datasets and unlearn labels involved in each unlearning scenario.

**Baselines:**   We implement the baselines with the following details.

*Retrain*: The dataset $\mathcal{D}^r$ is being divided among $K$ parties and the VFL model is retrained from scratch using the same hyperparameters as the baseline.

*Fine-Tuning* (Golatkar et al., 2020a; Jia et al., 2023): The dataset $\mathcal{D}^r$ is being divided among $K$ parties and the baseline VFL model is fine-tuned for 5 epochs with a learning rate of 0.01.

*Fisher Forgetting* (Golatkar et al., 2020a): The dataset $\mathcal{D}^r$ is being divided among $K$ parties, and the Fisher information matrix (FIM) is being used to inject Gaussian noise to perturb the VFL baseline models towards exact unlearning. For every backward propagation, once the gradient is calculated, each party inject Gaussian noise into their respective VFL baseline model to perturb the baseline model toward exact unlearning.

*Amnesiac* (Graves et al., 2021): The dataset is divided among $K$ parties and the baseline VFL model is retrained for 3 epochs, with the active party relabeling $y^u$ to an incorrect random label.

*Unsir* (Tarun et al., 2024): The dataset is being divided among $K$ parties. Each passive party introduces a noise matrix to the $\mathcal{D}_u$ image features they own. The noise added image features are used to impair each VFL baseline model, which is then repaired using the clean $\mathcal{D}^r$.

| Hyper-parameters | Single-label | | | | | | | | |
|---|---|---|---|---|---|---|---|---|---|
| | Resnet18-MNIST | Resnet18-CIFAR10 | Resnet18-CIFAR100 | Resnet18-ModelNet | Resnet18-Brain Tumor MRI | Resnet18-COVID-19 Radiography | Vgg16-CIFAR10 | Vgg16-CIFAR100 | MixText-Yahoo Answer |
| Optimization Method | SGD | SGD | SGD | SGD | SGD | SGD | SGD | SGD | SGD |
| Unlearning Rate | 2e-7 | 2e-7 | 5e-7 | 5e-7 | 6e-6 | 2e-7 | 2e-7 | 1e-7 | 7e-7 |
| Recovery Rate | 2e-7 | 2e-7 | 5e-7 | 5e-7 | 6e-6 | 2e-7 | 2e-7 | 1e-7 | 7e-7 |
| Unlearning Epochs | 10 | 20 | 10 | 4 | 4 | 5 | 17 | 9 | 30 |
| Number of $\mathcal{D}^{p,u}$ Data Samples | 40 | 40 | 30 | 30 | 15 | 40 | 40 | 30 | 30 |
| Number of $\mathcal{D}^{p,r}$ Data Samples | 3 | 3 | 3 | 3 | 3 | 3 | 3 | 3 | 3 |
| Batch Size | 32 | 32 | 32 | 32 | 32 | 32 | 32 | 32 | 32 |
| Weight Decay | 5e-4 | 5e-4 | 5e-4 | 5e-4 | 5e-4 | 5e-4 | 5e-4 | 5e-4 | 5e-4 |
| Momentum | 0.9 | 0.9 | 0.9 | 0.9 | 0.9 | 0.9 | 0.9 | 0.9 | 0.9 |

Table 9: Hyperparameters used for *single-label* unlearning in our proposed solution. In this single-label unlearning scenario, all datasets are processed using the same optimization method, stochastic gradient descent (SGD), with consistent parameters: a batch size of 32, a weight decay of 5e-4, and a momentum of 0.9. The recovery rate, which is the learning rate during the recovery phase, matches the unlearning rate across all datasets. Additionally, a minimal number of unlearning epochs (a maximum of 30) and unlearning data samples ($\mathcal{D}^{p,u}$, capped at 40) are utilized. Across all experiments, the number of recovery data samples ($\mathcal{D}^{p,r}$) used in the recovery phase is uniformly set to 3.

| Hyper-parameters | Two-label | | | | Multi-label | |
|---|---|---|---|---|---|---|
| | Resnet18-CIFAR10 | Resnet18-CIFAR100 | Vgg16-CIFAR10 | Vgg16-Cifar100 | Resnet18-CIFAR100 | Vgg16-CIFAR100 |
| Optimization Method | SGD | SGD | SGD | SGD | SGD | SGD |
| Unlearning Rate | 1e-6 | 9e-7 | 1e-6 | 9e-7 | 3e-6 | 1e-6 |
| Recovery Rate | 1e-6 | 9e-7 | 1e-6 | 9e-7 | 3e-6 | 1e-6 |
| Unlearning Epochs | 17 | 15 | 17 | 5 | 25 | 17 |
| Number of $\mathcal{D}^{p,u}$ Data Samples | 40 | 20 | 40 | 20 | 15 | 15 |
| Number of $\mathcal{D}^{p,r}$ Data Samples | 3 | 3 | 3 | 3 | 3 | 3 |
| Batch Size | 32 | 32 | 32 | 32 | 64 | 64 |
| Weight Decay | 5e-4 | 5e-4 | 5e-4 | 5e-4 | 5e-4 | 5e-4 |
| Momentum | 0.9 | 0.9 | 0.9 | 0.9 | 0.9 | 0.9 |

Table 10: Hyperparameters used for *two-label* and *multi-label* unlearning in our proposed solution. In these unlearning scenarios, all datasets are processed using the same optimization method, stochastic gradient descent (SGD), with consistent parameters: a weight decay of 5e-4, and a momentum of 0.9. For the two-label unlearning scenario, a batch size of 32 is used, while a batch size of 64 is employed for the multi-label unlearning scenario. The recovery rate, which is the learning rate during the recovery phase, matches the unlearning rate across all datasets. Additionally, a minimal number of unlearning epochs (a maximum of 25) and unlearning data samples ($\mathcal{D}^{p,u}$, capped at 40) are utilized. Across all experiments, the number of recovery data samples ($\mathcal{D}^{p,r}$) used in the recovery phase is uniformly set to 3.

*Boundary Unlearning* (Chen et al., 2023): The dataset $\mathcal{D}_u$ is being divided among $K$ parties. Passive parties create adversarial examples from the $\mathcal{D}_u$ image features they own, and the active party assigns a new nearest incorrect adversarial label to shrink the $\mathcal{D}_u$ to the nearest incorrect decision boundary.

*SSD* (Foster et al., 2024b): The dataset is divided among $K$ parties, and each party modifies their local model weights with the gradients of full data and $\mathcal{D}_u$.

**Datasets:** We conduct experiments on seven widely used datasets: MNIST, CIFAR10, CIFAR100, ModelNet, Brain Tumor MRI, COVID-19 Radiography and Yahoo Answers. For the MNIST, CIFAR10/100, Brain Tumor MRI and COVID-19 Radiography datasets. For all datasets, except ModelNet and Yahoo Answer, each image is split into two equal feature segments, with each segment assigned to a different passive party. For the ModelNet dataset, two 2D multi-view images are rendered for each 3D mesh model by placing virtual cameras at evenly spaced positions around the model's centroid. Each passive party is then assigned one of these rendered view. For Yahoo Answers dataset, each sample (*i.e.* a single paragraph of text) is splitted into two segments, with each passive party receiving one segment, ensuring that no single party has access to the complete text.

Figures 7 and 8 illustrate how image features of the dataset are being split among the passive party. In this setup, we consider a scenario involving two passive parties. For all image datasets, except ModelNet, each image's features are evenly split into two halves, as illustrated in Fig. 7. As an example in CIFAR10 (*i.e.*, class = bird), each passive party is assigned one half of the image (bird) features. This means that neither party has access to the full set of image features for any given data point, ensuring that the data remains fragmented and partially hidden from each individual party. This approach is specifically designed to enhance privacy and security by ensuring that no single party has access to the full information of the original image. This partitioning enables collaborative computation or machine learning tasks while maintaining data confidentiality.

| Scenarios | Models | Model of Passive Party | Model of Active Party | Datasets | Unlearn Labels |
|---|---|---|---|---|---|
| Single-label Unlearning | ResNet18 | 20 Conv | 1 FC | MNIST, CIFAR10, CIFAR100, ModelNet, COVID-19 Radiography | 0 |
| | ResNet18 | 20 Conv | 1 FC | Brain Tumor MRI | 2 |
| | Vgg16 | 13 Conv | 3 FC | CIFAR10, CIFAR100 | 0 |
| | MixText | 12 Brt | 4 FC | Yahoo Answer | 6 |
| Two-label Unlearning | ResNet18 | 20 Conv | 1 FC | CIFAR10, CIFAR100 | 0, 2 |
| | Vgg16 | 13 Conv | 3 FC | CIFAR10, CIFAR100 | 0, 2 |
| Multi-label Unlearning | ResNet18 | 20 Conv | 1 FC | CIFAR100 | 0, 2, 5, 7 |
| | Vgg16 | 13 Conv | 3 FC | CIFAR100 | 0, 2, 5, 7 |

Table 11: Models and datasets involved in each unlearning scenarios. FC: Fully-connected layer. Conv: Convolutional layer. Brt: Bert layer. In the single-label unlearning scenario, we perform unlearning for label *"0"* on the MNIST, CIFAR10, CIFAR100, ModelNet, and COVID-19 Radiography datasets using the ResNet18 architecture. Additionally, label *"0"* is unlearned on the CIFAR10 and CIFAR100 datasets using the Vgg16 architecture. For the Brain Tumor MRI dataset, we unlearn label *"2"* using the ResNet18 architecture. Similarly, on the Yahoo Answer dataset, label *"6"* is unlearned using the MixText architecture. In the two-label unlearning scenario, labels *"0"* and *"2"* are unlearned across all datasets and architectures. In the multi-label unlearning scenario, we extend the unlearning process to include labels *"0"*, *"2"*, *"5"*, and *"7"* across all datasets and architectures.

For the ModelNet dataset, a different approach is used to distribute data between the two passive parties as illustrated in Fig. 8. Specifically, each 3D mesh model is rendered into two distinct 2D images, each capturing a unique view of the object from different perspectives. Then, these two generated 2D views are assigned to separate passive parties, with each party receiving one view exclusively. As a result, no single party has access to both perspectives of the 3D mesh model, preserving information separation. This strategy not only safeguards data privacy but also enables collaborative analysis or training, ensuring that each party's contribution remains independent and incomplete.

Figure 9 illustrates how text features from the Yahoo Answers dataset are split between the passive parties. In this process, each paragraph is divided into two separate segments, with each segment representing a portion of the original text. Each passive party receives access to only one of these segments, ensuring that no single party can view the entire paragraph. This approach helps preserve the confidentiality of the dataset.

**MNIST** is a dataset of handwritten digits. MNIST dataset comprises 60,000 training examples and 10,000 test examples. Each example is represented as a single-channel image with dimensions of 28×28 pixels, categorized into one of 10 labels.

**CIFAR10/100** dataset contain 60,000 images (32×32 pixels, three color channels). CIFAR10 dataset includes 10 labels, with 50,000 for training and 10,000 for testing. The CIFAR100 dataset is similar but has 100 labels, each with 600 images, divided into 500 for training and 100 for testing.

**ModelNet** dataset is a widely-used 3D shape classification and retrieval benchmark, which contains 127,915 3D CAD models from 662 object categories.

**Brain Tumor MRI** is commonly used in healthcare scenarios. The Brain Tumor MRI dataset consists of 7,023 human brain MRI images categorized into four labels: Glioma, Meningioma, No Tumor, and Pituitary.

**COVID-19 Radiography** is a publicly available dataset on Kaggle, designed for research and development in medical imaging, specifically for detecting COVID-19 through chest X-rays. This dataset consists of chest X-ray images categorized into four classes, which are COVID, Normal, Viral Pneumonia and Lung Opacity.

**Yahoo Answers** is a dataset designed for text classification tasks, comprising 10 labels (topics) such as "Society & Culture", "Science & Mathematics", "Health", "Education & Reference", among others. Each label contains 140,000 training samples and 6,000 testing samples. For simplicity, we utilized 5,000 training samples and 2,000 testing samples from each label.

## A.9 LIMITATIONS

As the first method for label unlearning in VFL, our approach is developed under several standard and clearly defined assumptions. First, we assume access to a small public dataset that shares the same label space as the private data. This is an assumption commonly adopted in VFL and

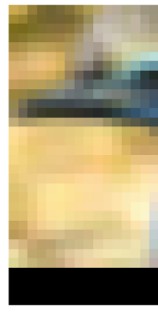 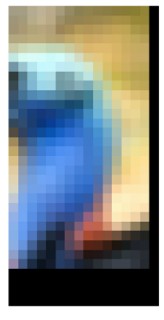

Passive Party A                Passive Party B

Figure 7: Illustration of how features from the CIFAR-10 dataset are split between passive parties. Passive Party A has access to one segment of the image, such as the left half of the original image's pixel data, while Passive Party B has access to the other segment, such as the right half. This division ensures that no single party has access to the complete image.

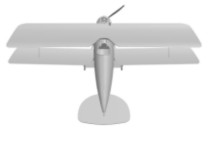 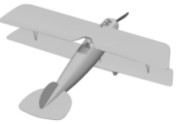

Passive Party A                Passive Party B

Figure 8: Illustration of how features from the ModelNet dataset are split between passive parties. Passive Party A is assigned one view of the 3D model, rendered from a specific angle (*e.g.*, an upper back-facing perspective of the airplane model), while Passive Party B receives a different view of the same model, rendered from another angle (*e.g.*, an upper right-side perspective). This setup ensures that no single party has full access to the complete 3D information.

can a loan corporation garnish your wages? yes they can read your contracts very carefully before signing also k

Passive Party A

now your rights as a consumer in your state go to your states website and search for the fconsumer rights section.

Passive Party B

Figure 9: Illustration of how text from the Yahoo Answers dataset is split between passive parties. Each passive party is assigned a different portion of the text, ensuring that no single party has access to the complete content. For example, the full paragraph is: *can a loan corporation garnish your wages? yes they can read your contracts very carefully before signing also know your rights as a consumer in your state go to your states website and search for the consumer rights section."* Passive Party A receives the first segment (*e.g.*, *can a loan corporation garnish your wages? yes they can read your contracts very carefully before signing also k"*), while Passive Party B receives the remaining portion (*e.g.*, *"now your rights as a consumer in your state go to your states website and search for the consumer rights section."*). This division maintains the confidentiality of the full paragraph.

vertical unlearning settings. When this label-space alignment holds, both our theoretical analysis and empirical results show that even a small number of public samples is sufficient to produce well-aligned gradient directions for effective unlearning. In extreme scenarios where the public set

is mislabeled or semantically unrelated, the public gradients may no longer reflect the true class boundary. Exploring more robust substitutes, such as pseudo-labels or synthetic anchor samples, represents an interesting direction for future work, particularly in sensitive domains like healthcare or finance, where suitable public analogues may be scarce.

Second, our experiments follow the synchronous VFL setting, which is the standard operational mode in existing vertical FL systems and in prior vertical unlearning studies. While our method is designed for this synchronous setting, extending it to asynchronous environments would require additional mechanisms for handling stale embeddings and misaligned updates.

Third, our method focuses on one-shot label unlearning rather than long sequences of deletion requests. In scenarios with many successive unlearning operations, accumulated updates may gradually shift the model.

Finally, our threat model assumes honest-but-curious participants and does not consider adversarial behaviors such as intentionally corrupted embeddings or poisoned gradients.

