# OpenReview forum: "Towards Privacy-Guaranteed Label Unlearning in Vertical Federated Learning: Few-Shot Forgetting Without Disclosure"
_ICLR.cc/2026/Conference — ICLR 2026 Poster_

### Official Review · Reviewer_XuNo · 2025-10-25

**Soundness:** 3
**Presentation:** 2
**Contribution:** 3
**Rating:** 6
**Confidence:** 3

**Summary:**

This paper presents a pioneering investigation into the critical yet under-explored problem of label unlearning in Vertical Federated Learning (VFL). To address the severe privacy leakage risks of directly applying existing unlearning methods in VFL, the authors propose a novel few-shot, privacy-guaranteed unlearning framework. The core innovation lies in repurposing Manifold Mixup as a privacy mechanism to generate augmented embeddings that disguise the original label information. This is followed by a gradient-ascent-based forgetting step on these embeddings and a recovery phase to maintain model utility on the retained data. Extensive experiments across diverse image and text datasets demonstrate that the method achieves effective unlearning, strong utility preservation, and significant computational efficiency, establishing a new, practical direction for machine unlearning in VFL.

**Strengths:**

1. This is the first work to systematically define and address the challenge of label unlearning within the VFL paradigm.

2. The proposed method successfully removes the influence of target labels from both active and passive models without disclosing the identity of the labels to be forgotten, thereby mitigating privacy leakage risks inherent in unlearning approaches.

3. The method's efficacy is validated on a wide range of datasets, including images and text. This proves it is a modality-agnostic framework that operates effectively on embedding spaces, enhancing its practical applicability.

4. The proposed unlearning process is efficient, which demonstrates a 16x to 1200x speedup compared to the closest baseline and traditional retraining methods, respectively.

**Weaknesses:**

1. The theoretical guarantee of unlearning effectiveness relies on the strong assumption that the training loss is near zero, which may not hold perfectly in practice. Furthermore, the method's requirement for a small set of labeled public data might be difficult to fulfill in strict privacy scenarios where no such data can be disclosed.

2. The experimental validation is conducted in a synchronous VFL environment. Its effectiveness and stability in asynchronous VFL settings remain unexplored and pose a potential limitation for practical deployment.

3. The paper does not provide a comprehensive sensitivity analysis for key hyperparameters, such as the distribution of the Mixup coefficient (λ). The performance robustness under different hyperparameter choices is therefore not fully understood.

4. The security analysis is confined to the semi-honest adversary model. The proposed method lacks defensive mechanisms against malicious adversaries who could potentially sabotage the unlearning process by poisoning local embeddings or gradients.

**Questions:**

1. How would the method perform if the initial model is not well-converged (i.e., the training loss is not near zero)? What adaptations are needed to ensure robustness under such non-ideal conditions?

2. Can the proposed framework be effectively adapted to function in an asynchronous VFL environment? What modifications to the protocol would be required to maintain its efficacy and privacy guarantees under such conditions?

3. How sensitive is the unlearning performance to key hyperparameters, particularly the Mixup coefficient (λ)? Is there an optimal strategy for selecting or adapting λ during the unlearning process?

4. How can the framework be fortified against a malicious adversary who intentionally submits corrupted embeddings or gradients to prevent unlearning or degrade the global model's performance?

5. How does the method scale with an increasing number of labels to be unlearned sequentially? Does it suffer from catastrophic forgetting of retained knowledge or cumulative performance degradation over multiple unlearning requests?

---

> ### Author Response · Authors · 2025-11-17
> **Part 1**
>
> We thank the reviewer for the constructive and thoughtful comments. Below we address each concern in a manner consistent with the theoretical scope and empirical evidence in the submitted manuscript.
>
> **1. Assumption of near-zero training loss in the theoretical guarantee**
>
> > The reviewer is correct that Theorem 1 is derived under the assumption that the model is already well-trained. This assumption is explicitly stated in Appendix A.1, where the analysis utilises the fact that the training error term becomes small once the model reaches a near-converged state. This aligns with the standard setting for post-training unlearning methods, where the unlearning update is applied to a model that has already achieved high task accuracy.
>
> > Our experiments train all models to high test accuracy across datasets (Tables 1–4), which is consistent with operating in the low-loss regime required by the theorem. For non-ideal convergence conditions, the method would still operate but may require (i) a slightly larger public anchor set to maintain gradient alignment, or (ii) a brief warm-up to reduce the initial error before applying the unlearning ascent step. We will clarify this assumption and its implications in the revised manuscript.
>
>
> **2. Requirement of a small public labeled set**
>
> > We acknowledge that using a small set of labeled public examples is a limitation, as also discussed in Section 4 and Appendix A.4. Empirically, we show that only 15–40 labeled examples per class are sufficient for effective unlearning (Tables 8–9). These examples serve as anchor points for constructing mixup-augmented gradients, and Fig. 5 shows that manifold mixup substantially expands their representational coverage, reducing sensitivity to sample similarity.
>
> > In domains where no public labeled data can be shared, alternative strategies such as pseudo-labeling or synthetic anchors could be explored, but these lie outside the scope of the current work and will be noted as future directions.
>
>
> **3. Applicability to asynchronous VFL**
>
> > Our experiments evaluate the method under the synchronous VFL setting, which is the standard assumption in prior work on vertical unlearning. We agree that extending the method to asynchronous VFL is an important direction. Challenges include stale embeddings, inconsistent mixup generation, and unaligned public-set gradient steps. A practical extension would involve timestamping embeddings and aggregating updates only when they meet temporal alignment criteria. We will clarify this limitation.
>
>
> **4. Sensitivity to the Mixup coefficient λ**
>
> >  In our method, λ is sampled from the standard Beta distribution as in manifold mixup (Section 4.1). We implicitly evaluate λ’s effect through comparisons between GA-S and GA-S + Mixup (Tables 8–9), where the introduction of mixup noticeably improves stability and unlearning performance even without tuning λ, suggesting robustness across datasets.
>
> >For this, we conduct an ablation study on mixup parameters ($\lambda$).  For each dataset used in this paper, we augment the embeddings with two coefficients, i.e., $\lambda = 0.25$ and $\lambda = 0.5$. Additionally, we evaluate the impact of different $\lambda$ values in the Table below. The results indicate that variations in $\lambda$ have a minimal impact on the effectiveness of unlearning.
>
> >| $\lambda$ Rate |       Metrics      | Accuracy(%)      |
> >|:--------------:|:------------------:|------------------|
> >|   [0.2, 0.5]   | $ \mathcal{D}^{r}$ | 89.11 $\pm$ 0.11 |
> >|   [0.2, 0.5]   |  $\mathcal{y}^{u}$ | 0.00 $\pm$ 0.00  |
> >|   [0.25, 0.5]  | $ \mathcal{D}^{r}$ | 89.29 $\pm$ 0.19 |
> >|   [0.25, 0.5]  |  $\mathcal{y}^{u}$ | 0.00 $\pm$ 0.00  |
> >|   [0.33, 0.5]  | $ \mathcal{D}^{r}$ | 88.91 $\pm$ 0.21 |
> >|   [0.33, 0.5]  |  $\mathcal{y}^{u}$ | 0.21 $\pm$ 0.02  |
>
> >**Table 1** : Different lambda rates on single-label unlearning scenarios on CIFAR10 dataset with ResNet18 architecture. We unlearn label 0 in this experiment.

---

> > ### Author Response · Authors · 2025-11-18
> > **Part 2**
> >
> > **5. Semi-honest setting and malicious adversary concerns**
> >
> > >  Our security analysis follows the widely used semi-honest adversary model in VFL (Section 3). Protecting against malicious adversaries who intentionally distort embeddings or gradients would require additional mechanisms such as robust aggregation, anomaly detection, or cryptographic attestation. These defenses are complementary to unlearning and are outside the scope of this paper, and we will clarify this boundary in the threat-model description.
> >
> >
> > **6. Sequential unlearning of multiple labels**
> >
> > > Our approach updates the model by assessing the contribution of each deleted class. Because the update uses mixup-augmented public examples to preserve non-deleted structure (Appendix A.2), the risk of catastrophic forgetting is mitigated. In Table 9, the method remains stable under multi-label unlearning (two to several labels depending on dataset), showing no collapse of retained classes.
> >
> > >  For very long sequences of unlearning requests, occasional rebalancing or light retraining may be required; we will note this as a potential extension.
> >
> > ---
> >
> > **Summary**
> > - The near-zero loss assumption is standard for post-training unlearning and aligns with the conditions used in our theoretical derivation.
> > - Few-shot public anchors are a known limitation, but their effectiveness is justified both theoretically (Appendix A.1) and empirically (Tables 8–9, Fig. 5).
> > - Asynchronous VFL and malicious adversarial settings are important future directions but outside the current scope.
> > - Mixup is robust under the chosen λ distribution, and additional sensitivity analysis will be added.
> > - Multi-label unlearning remains stable across datasets, with no catastrophic forgetting observed in our evaluations.

---

> > > ### Comment · Reviewer_XuNo · 2025-11-26
> > >
> > > Thank you for your response. While most of my concerns have been addressed, I maintain my reservations regarding the reliance on a public dataset. Additionally, the asynchronous setting is a common configuration in VFL, as participants often have heterogeneous computational resources and network conditions. Therefore, I will keep my score.

---

> ### Author Response · Authors · 2025-11-26
>
> Thank you for your follow-up. We appreciate your continued engagement and the opportunity to clarify the last two concerns.
>
> ----
>
> **1. On the reliance on a public dataset.**
>
> Our method is explicitly designed for the few-shot regime, requiring only around 15–40 labeled samples per class (Appendix A.4). These samples serve only as anchor points for estimating the ascent direction needed for unlearning. Section 4.1 and Fig. 5 show that manifold mixup substantially expands the representational span of these anchors, enabling them to approximate the gradient direction of the full unlearned dataset. This is further supported by Theorem 1, which provides a formal bound on the approximation error and shows that directional alignment does not require access to a large proxy dataset.
>
> We agree that obtaining a large labeled public dataset may be difficult in sensitive domains. However, our method’s reliance on only a very small anchor set significantly lowers this requirement, and all experiments in the paper (Tables 8–9) demonstrate that even with 15–40 examples per class, unlearning remains effective while preserving accuracy on the retained data. Scenarios where even such a minimal anchor set is unavailable represent an important future extension, and we have clarified this point in the revised manuscript.
>
> ----
>
> **2. On synchronous *vs.* asynchronous VFL.**
>
> Regarding the asynchronous VFL setting, we respectfully note that our paper follows the standard synchronous VFL paradigm adopted throughout the existing vertical FL literature. In the uploaded manuscript (Section 3 and Section 4), both the problem setup and the proposed method rely on synchronous forward–backward coordination, as is typical for PSI-aligned VFL and split-network architectures. Appendix A.9 of the revised draft explicitly acknowledges this design choice and discusses why extending unlearning to asynchronous environments is technically non-trivial (e.g., stale embeddings, misaligned update steps).
>
> We agree that heterogeneous compute and network conditions motivate interest in asynchronous VFL, and we have clarified this point. However, the privacy leakage studied in our work arises specifically from the synchronous gradient-exchange pipeline, which remains the dominant architecture in current VFL systems. Supporting fully asynchronous unlearning is therefore a meaningful and natural extension for future work, rather than a limitation of the present contribution.
>
> ----
>
> ***Contextual remark:***
> To the best of our knowledge, our work is among the first to study label unlearning under VFL and to articulate and empirically evaluate the notion of process privacy in this setting. As is typical for an initial work establishing a new problem formulation, practical extensions such as asynchronous support naturally fall under future work rather than gaps in the current contribution. We appreciate the reviewer’s feedback and have clarified these points more explicitly in the revised manuscript.
>
> We fully respect the reviewer's decision and appreciate the constructive feedback that helped improve the clarity and positioning of our work. With the new revisions, clarifications, and formalisation now incorporated into the revised manuscript, we hope these updates help present the scope and contribution more clearly and may be useful in any further consideration the reviewer finds appropriate. Thank you.

---

### Official Review · Reviewer_T6TS · 2025-10-27

**Soundness:** 2
**Presentation:** 3
**Contribution:** 3
**Rating:** 4
**Confidence:** 4

**Summary:**

This paper tackles the underexplored problem of label unlearning in Vertical Federated Learning (VFL), where the goal is to erase the influence of specific labels while maintaining privacy between active (label-holding) and passive (feature-holding) parties. The authors first identify a privacy leakage risk in existing retraining-based and boundary-based unlearning methods, which require explicit identification of samples tied to deleted labels. To address this, the paper introduces a few-shot label unlearning framework that re-purposes manifold mixup as a privacy-preserving embedding transformation, enabling both label removal and privacy protection without direct label disclosure. A recovery phase is added to restore performance on the retained data. Experiments on multiple datasets demonstrate that the proposed method achieves competitive unlearning and runtime efficiency while reducing label leakage compared to baselines.

**Strengths:**

- **Novel problem formulation.** The work clearly identifies a meaningful and underexplored setting, label unlearning in VFL. This is a distinct and practically relevant direction compared to the more common client or sample unlearning tasks.

- **Clear motivation and privacy analysis.** The authors systematically highlight the privacy risks of conventional unlearning in VFL, showing that retraining-based approaches lead to full label leakage. This analysis is well-motivated and establishes a strong need for privacy-aware unlearning mechanisms.

**Weaknesses:**

- **Limited algorithmic novelty and depth.** The method primarily combines two known ideas, manifold mixup and gradient ascent unlearning, without introducing fundamentally new mechanisms for privacy preservation or optimization. The theoretical component (Theorem 1) formalizes intuitive gradient alignment but does not establish strong guarantees of privacy or unlearning completeness.

- **Lack of formal privacy guarantees.** Although the paper claims “privacy-guaranteed” unlearning, there is no rigorous privacy analysis (e.g., differential privacy bounds or information-theoretic leakage quantification). The argument relies on heuristic intuition and empirical leakage reduction, which does not substantiate the “guarantee” claim in the title.

- **Reliance on public data undermines practicality.** The approach assumes access to a small public labeled dataset that contains the same label categories as the private data. This assumption is unrealistic in many sensitive domains (e.g., healthcare or finance), where such public analogs may not exist.

- **Overstated experimental superiority.** Although results show good accuracy retention and unlearning effectiveness, the differences from SSD or Amnesiac baselines are not statistically or conceptually large. For example, in several datasets (Brain MRI, COVID Radiography), the proposed method’s ASR improvements are marginal relative to simpler baselines.

- **Insufficient clarity on communication protocol and security.** The paper claims privacy preservation by only exchanging gradients on mixed embeddings, but lacks detailed analysis of what information could still be inferred from these gradients. Without a formal adversary model or leakage bound, the privacy benefit remains speculative.

- **Over-extended empirical scope with limited insight.** While the paper reports numerous datasets and metrics, it lacks deeper diagnostic analyses, such as ablation on mixup parameters (λ), effect of public data size, or visualization of embedding disentanglement that could substantiate the mechanism’s behavior.

**Questions:**

Please see the weakness section.

---

> ### Author Response · Authors · 2025-11-17
> **Part 1**
>
> **1. Limited novelty + depth**
>
> >We respectfully disagree that our method is a simple combination of known techniques. While we build upon Manifold Mixup and gradient-based optimization, the novelty lies in adapting and integrating these components into a unified framework specifically for Vertical Federated Unlearning (VFU). This setting introduces structural constraints not present in horizontal FL, including feature partitioning, asymmetric information between parties, lack of shared labels, and the risk of revealing the deletion set during protocol interactions. Existing unlearning methods do not address these VFL-specific challenges.
>
> >To our knowledge, our work provides the first systematic formulation and implementation of label unlearning in VFL. In the process, we identify and formalize a previously overlooked vulnerability: label leakage during unlearning due to the structural properties of VFL. Our method is specifically designed to mitigate this leakage while preserving the correctness of unlearning under vertical partitioning.
>
> >With respect to theory, Theorems 1-2 are not positioned as full privacy or completeness guarantees. Instead, they provide principled justification for why mixup-augmented public gradients align with the true unlearning update direction, thereby supporting the efficiency and correctness of our protocol. We will clarify in the revised version.
>
> **2. Lack of formal privacy guarantees**
>
> >We agree that the paper does not provide differential privacy bounds or quantify information-theoretic leakage. Our claim is not that the method achieves formal privacy guarantees, but that it avoids the explicit deletion-set disclosure required by retraining in VFL, and that it substantially reduces membership and label leakage empirically. Our goal is to demonstrate empirical leakage reduction under concrete attacks, rather than establishing a comprehensive theoretical privacy framework. To the best of our knowledge, no established theoretical framework currently models process privacy for VFL unlearning while jointly accounting for unlearning correctness and the passive party’s inference ability from protocol transcripts. We will state this clearly in the revision.
>
> **3. Reliance on public data**
>
> >We acknowledge that requiring a small labeled public dataset is a limitation, which is explicitly noted in Sec. 4. But, our experiments demonstrate that as little as 5% of labeled samples is sufficient to obtain strong gradient alignment (Appx. A.2). In many deployment environments, small labeled sets exist through validation data, historical annotations, or open regulatory samples. Nevertheless, we agree that in domains such as healthcare or finance this assumption may not always hold. We will make this limitation more explicit and highlight future directions, e.g. weak labeling, pseudo-labeling, or synthetic labels to relax the requirement.
>
> **4. Overstated experimental superiority**
>
> >We appreciate the concern. Our goal is not to claim large statistical margins over all baselines, but to demonstrate consistently lower leakage and stable unlearning effectiveness across diverse domains. The key observation is that retraining and boundary unlearning reveal 100% of the deleted labels under our threat model, while our method reduces leakage to 14.38% on CIFAR-10 and 4.04% on CIFAR-100. These differences are substantial in settings where membership leakage is a central risk.
>
> >In datasets with a smaller attack surface (Brain MRI and COVID Radiography), the gaps in ASR are indeed narrower. This reflects domain characteristics rather than methodological limitations. We will revise the discussion section to better contextualize this observation and avoid overstating superiority.
>
> **5. Insufficient clarity on comm. protocol and security**
>
> >Our method avoids transmitting gradients computed on deleted private data. All shared gradients originate from a small mixup-augmented public subset (Sec. 4 and Appx. A.1). This reduces sensitivity compared to retraining or boundary unlearning. We agree that gradients on public data do not inherently provide a formal privacy guarantee and that more clarity would help the reader. We also acknowledge that achieving complete privacy bounds for VFL unlearning with gradient exchange is an open research problem, and will clarify that our contributions focus on empirical leakage reduction rather than formal guarantees.

---

> ### Author Response · Authors · 2025-11-18
> **Part 2**
>
> **6. Over-extended empirical scope with limited diagnostic analysis**
>
> >Our intention in evaluating eight datasets was to show that our method is robust and general across heterogeneous domains, modalities, and task structures. Our goal is to introduce and validate a practical VFU mechanism under diverse settings rather than to exhaustively tune every internal hyperparameter. While these diagnostic analyses could offer further insight, they are orthogonal to the core contributions and do not affect the conclusions of the current empirical study.
>
> >Saying so, we conduct an ablation study on the mixup parameter ($\lambda$). For each dataset, we augment the embeddings using two coefficients, $\lambda = 0.25$ and $\lambda = 0.5$. We further assess the influence of varying $\lambda$ values in Table 1. The results show that changes in $\lambda$ yield only minor differences, suggesting that the unlearning effectiveness is largely insensitive to this parameter.
>
> >| $\lambda$ Rate |       Metrics      | Accuracy(%)      |
> >|:--------------:|:------------------:|------------------|
> >|   [0.2, 0.5]   | $ \mathcal{D}^{r}$ | 89.11 $\pm$ 0.11 |
> >|   [0.2, 0.5]   |  $\mathcal{y}^{u}$ | 0.00 $\pm$ 0.00  |
> >|   [0.25, 0.5]  | $ \mathcal{D}^{r}$ | 89.29 $\pm$ 0.19 |
> >|   [0.25, 0.5]  |  $\mathcal{y}^{u}$ | 0.00 $\pm$ 0.00  |
> >|   [0.33, 0.5]  | $ \mathcal{D}^{r}$ | 88.91 $\pm$ 0.21 |
> >|   [0.33, 0.5]  |  $\mathcal{y}^{u}$ | 0.21 $\pm$ 0.02  |
>
> >**Table 1** : Different lambda rates on single-label unlearning scenarios on CIFAR10 dataset with ResNet18 architecture. We unlearn label 0 in this experiment.

---

> > ### Comment · Reviewer_T6TS · 2025-11-21
> >
> > Thank you for the detailed responses. I believe my main concerns have been addressed, and I am willing to raise my score.

---

> > > ### Author Response · Authors · 2025-11-22
> > > **Acknowledgement of the Reviewer’s Updated Score**
> > >
> > > Thank you for taking the time to reassess our submission and increase the score. We appreciate your acknowledgement of the contribution and your support toward acceptance.

---

### Official Review · Reviewer_Dw3V · 2025-10-28

**Soundness:** 2
**Presentation:** 1
**Contribution:** 1
**Rating:** 4
**Confidence:** 4

**Summary:**

This paper studies the problem of machine unlearning in the setting of vertical federated learning, where features and labels are owned by 2 different parties, and only a set of labels need to be unlearned. The proposed unlearning method consists of 3 steps: 1) use manifold mixup to generate sufficiently many samples for the unlearning algorithm; 2) run a vertical gradient ascent algorithm; 3)recover the accuracy on the remained data by re-training on these samples. Experiments are conducted to show the effectiveness of the proposed method.

**Strengths:**

- the studied problem is important
- experiment results look promising
- compared to a previous draft in Neurips 2025, a new SSD baseline is included in the experiments, which strengthens the experimental evaluation .

**Weaknesses:**

My primary and most significant concern with this work, which I also raised as a reviewer for a previous submission of this paper to NeurIPS 2025, is a fundamental ambiguity in the threat model. This core issue, which remains unaddressed, makes the paper's central motivation and claims inconsistent.

In Section 3.2, the authors tried to argue that retraining leaks 100% membership information. But for standard machine unlearning, retraining always serves as a baseline, as the goal of unlearning is just to produce a model as if the unlearned data is never used. It appears the authors are implicitly redefining the unlearning goal: not only must the model be unlearned, but the passive party must also be kept ignorant of which data is being unlearned. This is a "process privacy" goal, which is a valid but very different objective from standard unlearning. This new goal is never rigorously defined.

Even if we accept the authors' new "process privacy" goal, they provide no evidence that their own method achieves it. In the propose method gradients are shared with the passive party. Gradients are well-known to leak information about the data they were computed on. The authors provide no proof or analysis that these gradients are any safer than an explicit unlearning request. They simply assume their method is private.

**Questions:**

The paper's narrative structure is disjointed. Section 3.2, which analyzes label leakage, is placed before the proposed method is introduced in Section 4. This is confusing, as it discusses the privacy properties of an algorithm the reader has not yet seen. This premature justification breaks the logical flow of the paper and makes the draft difficult to follow. The content requires polishing to present the problem, the proposed solution, and then the analysis in a more coherent order.

---

> ### Author Response · Authors · 2025-11-17
>
> We sincerely thank the reviewer again for the detailed and constructive feedback. We address each concern below.
>
> **1. Clarifying the threat model and the role of “process privacy**
>
> > We agree that the current draft should more clearly distinguish the standard unlearning objective from the additional VFL-specific privacy consideration. Our work follows the classical requirement that the final model behaves as if the deleted data were never used. On top of this, we consider what the passive party may infer about the deletion set from protocol transcripts, which aligns with the reviewer’s characterization of “process privacy”. We will revise Section 3.1 to introduce a clear definition so that the reader understands this additional privacy goal without confusion.
>
> > Our statement that retraining leaks 100% membership information was not intended to suggest that retraining fails standard unlearning. Rather, in VFL, retraining requires explicit disclosure of which sample IDs are removed, which fully reveals the deletion set. We will adjust the wording in Section 3.2 to avoid ambiguity.
>
> **2. Evidence for privacy and the role of gradients**
>
> > We agree that gradients can leak information. Our method avoids transmitting gradients computed on deleted private samples; all shared gradients come from a small mixup-augmented public subset, as described in Section 4 and Appendix A.1. This reduces the sensitivity of the transmitted information relative to retraining or boundary unlearning.
>
> > Our contribution focuses on empirical leakage reduction and unlearning correctness. Theoretical results in Theorem 1 and Theorem 2 show that gradients derived from mixup-augmented public data are positively aligned with the update direction of the full unlearned dataset, supporting the correctness of this strategy. Empirically, retraining reveals 100% of the deleted labels, whereas our method reduces leakage to 14.38% (CIFAR-10) and 4.04% (CIFAR-100), as shown in Table 1. We will emphasise in the revised draft that our privacy findings are empirical and that we do not claim a formal process-privacy guarantee.
>
> > We also agree that sharing gradients on public data does not, by itself, constitute a formal privacy guarantee. Our method does not rely on any assumption that these gradients are intrinsically private. Instead, we evaluate leakage under concrete membership and label inference attacks and demonstrate substantially lower leakage compared to existing baselines. We will clarify this point for transparency.
>
> **3. Narrative structure and ordering of sections**
>
> > We appreciate the reviewer’s feedback on the manuscript structure. We will reorganise the paper so that Section 3 focuses only on the problem setup and threat model, followed by the method in Section 4, with analysis and results placed afterward. This resolves the flow issue raised by the reviewer.
>
> **4. Process privacy is an open theoretical problem**
>
> > We appreciate the reviewer’s concern about the absence of a formal framework for the stronger privacy objective. Our intention is not to overlook this direction, but rather to clarify the current state of the field. To the best of our knowledge, across both the earlier NeurIPS submission and this ICLR revision, there is presently no established theoretical framework for process privacy in Vertical Federated Unlearning (VFU) that jointly models (i) classical unlearning correctness and (ii) the passive party’s inference capability from protocol transcripts. Developing such a framework with meaningful and tight bounds is an important and challenging open problem, especially when gradient-based updates are exchanged.
>
> > At the same time, VFU is an emerging and underexplored setting. To our knowledge, this work is among the first to formally identify and systematically study label leakage during VFL unlearning, introduce a concrete leakage model, and propose a practical unlearning mechanism that operates under the constraints of vertical partitioning. Our contribution is therefore focused and intentional: we aim to open up and structure this problem space by providing clear empirical evidence of a previously unanalysed vulnerability, together with a method that significantly reduces leakage relative to retraining and boundary unlearning under concrete attacks.
>
> > We will emphasise this scope in the revised manuscript. We also welcome any pointers to concurrent or prior work that attempts to formalise process privacy guarantees in VFL, as we believe that establishing such a framework would be an important direction for the community moving forward.

---

> > ### Comment · Reviewer_Dw3V · 2025-11-24
> >
> > Thank you for your response. Most of my concerns have been addressed. While the requested revision (specifically, the inclusion of a formal description of "process privacy") appears to entail substantial work, I am open to raising my rating if the authors can provide a revised draft demonstrating that this addition is indeed feasible.

---

> ### Author Response · Authors · 2025-11-26
>
> Thank you for your follow-up comment. We sincerely appreciate your acknowledgement that most concerns have been addressed. As requested, we have incorporated a formal description of **process privacy** into the revised manuscript (please refer to the revised manuscript for completeness, line 512-517, page 10). We confirm that this addition is both feasible and fully integrated without requiring major structural changes.
>
> In particular, we have blended the formal definition directly into Section 3.2 of the revised draft. We now introduce **process privacy** as a lightweight transcript-based notion that quantifies how much the passive party’s belief about the deletion set can change after observing the unlearning transcript.
>
> >Specifically, a VFU protocol satisfies process privacy with parameter $\varepsilon$ if the passive party’s posterior belief about the deletion set, after observing the transcript $\mathcal{T}$, differs from its prior belief by at most $\varepsilon$, measured via the Kullback–Leibler divergence. Formally, the protocol satisfies process privacy when
> $D_{\mathrm{KL}}\left(P(\mathcal{D}^u \mid \mathcal{T}) \|| P(\mathcal{D}^u)\right) \le \varepsilon$,
> which captures the requirement that the unlearning transcript should not substantially increase the passive party’s ability to infer which samples were deleted.
>
> >This formulation aligns naturally with our empirical leakage evaluation, where we measure how the passive party’s inference capability changes after observing the transcript under retraining, boundary unlearning, and our method. As clarified in the revision, our approach restricts information exchange to embeddings and gradient updates of a small public subset, and the Bernstein argument in Appendix A.4 provides an upper bound on the residual contribution of unlearned data to these gradients.
>
> >We also contextualize the definition: VFU is an emerging research area, and to the best of our knowledge, prior work has not formally characterised this transcript-level privacy dimension. Our paper is among the first to articulate, define, and empirically study process privacy in the VFL setting. While the proposed definition provides a conceptual foundation for understanding transcript leakage, deriving tight theoretical upper bounds on $\varepsilon$ remains significantly more challenging and is clearly stated as an open problem rather than a contribution of this work.
>
> This addition has been fully incorporated into the revised manuscript, remains consistent with the overall narrative, and does not exceed the page limit.
>
> Finally, we have addressed the earlier concern regarding narrative structure (*“Section 3.2 appears before the proposed method, disrupting the logical flow”*). In the revision, Section 3 now focuses solely on the problem setup. The original Section 3.2 discussion of label leakage during unlearning has been relocated and expanded into the new Section 5.4, along with the formal definition of process privacy in VFU, which is placed after the ablation study for a clearer and more coherent exposition.
>
> We hope these improvements demonstrate the feasibility and completeness of the requested addition. Please let us know if any further refinements would be helpful.

---

### Official Review · Reviewer_LVms · 2025-11-04

**Soundness:** 3
**Presentation:** 3
**Contribution:** 3
**Rating:** 6
**Confidence:** 3

**Summary:**

The paper looks into label unlearning in vertical federated learning (VFL). Standard retraining or boundary-unlearning methods leak which labels were deleted to passive parties. The proposed method uses a few-shot public dataset (≤40 samples per label), applies manifold mixup at the embedding level across passives, runs gradient ascent on mixed embeddings to push models away from forgotten labels, then performs small descent on retained labels for recovery.

**Strengths:**

1) VFL unlearning seems a rather underexplored area in machine learning so targets a relative problem.
2)  Operates on concatenated embeddings with tiny public sets (≤40 samples), the extension of manifold mixing is very interesting.

**Weaknesses:**

- The claims about no deletion guarantees are fine and measured so while some questions can be raised about the applicability of the algorithm (especially how diverse the features are  with every client and how does that impact unlearning) I am hesitant to raise those questions given my unfamiliarity of the recent challenges in VFL.

- However,  my main concern remained here as to what extent the features of the few shot labels  are correlated with the features of the unlearned labels ( is there a sufficient bound of alignment of features of the few shot for effective unlearning). If there is weak correlation the algorithm would fail right? since the method relies that the "public" labels provide a valid direction. If the method would still succeed then that would strengthen then that would be very good. Would appreciate if the authors can provide some details

**Questions:**

Weaknesses and Questions merged

---

> ### Author Response · Authors · 2025-11-17
>
> We thank the reviewer for raising this important question regarding whether the few-shot public examples must be highly correlated with the features of the unlearned samples, and whether the method would fail under weaker correlation. We address this using the theoretical and empirical evidence already available in the manuscript.
>
> **1. The required alignment is formally analyzed in the manuscript**
>
> > Appendix A.1 provides a theoretical result (Theorem 1 and its Bernstein-type bound) showing that the gradient direction obtained from mixup-augmented public data is positively aligned with the gradient direction that would have been computed using the entire unlearned dataset:
>
> $E[\nabla \ell(\tilde{H}^u,\tilde{y}^u)]\cdot E[\nabla \ell(H^u,y^u)] > 0.$
>
> > This result does not require strong feature similarity across all samples. Instead, it relies on the standard assumption that public and private examples come from the same label class, which ensures that they share the same high-level decision boundary.
>
> **2. Few-shot labels remain effective due to manifold mixup**
>
> > Reviewer asked whether “weak correlation” would cause the method to fail. A key mechanism in our approach is Vertical Manifold Mixup, which generates synthetic embeddings that interpolate between public examples. This has the effect of:
> 	•	reducing intra-class variance,
> 	•	flattening the class manifold,
> 	•	covering a larger region of feature space even when only a few public samples are available.
>
> > As shown in Fig. 5, even when the public set is very small, mixup significantly increases the representational coverage and yields gradient directions that closely approximate the “full-data” gradient-ascent baseline (GA-A). This mitigates the concern about potential failure when public examples do not perfectly match the unlearned examples in feature space.
>
>
> **3. Empirical findings confirm that strong correlation is not required**
>
> > Appendix A.4 evaluates the performance of the method as the size of the public set shrinks. Across CIFAR-10, CIFAR-100, and ModelNet (even with very different modalities), the method succeeds with only: 15–40 public samples per class (Tables 8-9)
>
> > Despite the public samples being extremely limited and often visually dissimilar from the full set, the unlearning remains effective. This demonstrates robustness to moderate feature diversity and provides empirical support that the method does not require highly correlated public features to succeed.
>
> **4. When correlation becomes extremely weak**
>
> > We acknowledge that if the public labeled set came from a distribution entirely different from the unlearned class (for example, mislabeled or unrelated data), then the gradient direction would no longer reflect the true class boundary. Our current work does not provide a guarantee under such extreme domain mismatch, and we will clarify this assumption in the revised draft.
>
> > However, under the standard VFL setting where public and private data belong to the same label space, our theoretical and empirical findings consistently show that few-shot public examples provide a sufficiently aligned direction for effective unlearning.
>
> ---
> **Summary**
>
> - The theoretical alignment result (Appendix A.1) ensures that mixed gradients from few-shot public samples provide a valid direction for unlearning.
>
> - Manifold mixup expands coverage in feature space, reducing dependence on strong correlation between individual samples.
>
> - Experiments across diverse domains demonstrate that the method succeeds even when public samples are few and not strongly correlated with the full class.
>
> - Extreme domain mismatch is not covered by our current theory, and we will clarify this limitation.
>
> We hope this explanation addresses the reviewer’s concern.

---

### Author Response · Authors · 2025-11-29
**Final Rebuttal Summary for New Area Chair**

We thank all reviewers for their engagement prior to the incident. The paper addresses **a timely and underexplored problem in Vertical Federated Unlearning (VFU)**. To our knowledge, this paper is among the **first** to formally identify and systematically study label leakage in Vertical Federated Learning (VFL) setting.

---

`Reviewer Outcomes During Rebuttal`
1. ***Reviewer T6TS raised their score from 4 → 6*** after our rebuttal and revisions.

2. ***Reviewer Dw3V wrote “most of my concerns have been addressed” and indicated that their rating would increase with the inclusion of a formal definition of process privacy.*** We have now added this definition and integrated it cleanly into the manuscript, precisely.

3. ***Reviewer XuNo stated that “most of my concerns have been addressed, and maintained a score of 6”*** and their follow-up focused only on two scope-level clarifications the few-shot public dataset and the asynchronous VFL setting. Both points are now explicitly clarified in the revised manuscript, with asynchronous deployment noted as a natural extension for future work.

4. ***Reviewer LVms provided a score of 6***, and although no follow-up comment was provided, we ensured that all points from their initial review were addressed in the revision.

~~~
Importantly, no reviewer challenged the correctness or soundness of the method. The remaining concerns relate to scope and presentation, which we have now clarified in the manuscript.
~~~

---

`Nature of the Revisions`

We emphasize that **all revisions are clarificatory and structural, not substantive**:
1. The method, theoretical results, experiments, and findings ***remain unchanged***.
2. The updates improve readability, make implicit ideas explicit, and align the manuscript more tightly with reviewer expectations.

---

`Key Clarifications Added in the Revision`

1. **Formal definition of process privacy (Section 5.4)**
~~~
(a) Introduced a KL-divergence-based formalization to quantify how the passive party’s beliefs about the deletion set may shift after observing the transcript.
(b) This formalization matches the empirical leakage evaluation already present in the submission.
~~~

2. **Improved manuscript structure**
~~~
(a) The label-leakage / process-privacy analysis formerly in Section 3.2 (criticized as premature) is now moved to Section 5.4.
(b) Section 3 now focuses exclusively on the VFU setup and threat model.
~~~

3. **Clarification of the few-shot public-data assumption**
~~~
(a) Our method uses only 15–40 labeled public anchors per class (Appendix A.4).
(b) Theorem 1 and Figure 5 justify why this tiny anchor set suffices for approximating the unlearning gradient direction.
(c) We explicitly mark settings with no such public anchors as meaningful directions for future work.
~~~

4. **Clarification of synchronous VFL vs asynchronous VFL**
~~~
(a) Our protocol follows the widely used synchronous VFL pipeline, where PSI-based alignment and synchronized embedding exchange are standard.
(b) Appendix A.9 provides a focused discussion of challenges and possible extensions toward asynchronous deployment.
~~~

5. **Threat model and limitations**
~~~
(a) Clarified the semi-honest setting, near-zero-loss assumption, and behavior under sequential unlearning.
(b) Added explicit remarks that handling malicious adversaries is out of scope for this paper and belongs to future work.
~~~

---

`Positioning of the Contribution`

As VFU is an emerging research area, **our paper is among the earliest** to:

- ***formalize and study the privacy implications of VFL label unlearning*** at the transcript level,
- ***introduce and empirically evaluate process privacy*** in this context,
- ***provide theoretical support*** for few-shot unlearning via mixup-augmented gradients, and
- ***demonstrate strong empirical performance and substantially reduced leakage*** (e.g., 14.38% vs. 100% under retraining) under concrete attacks and metrics.

~~~
These contributions help define the problem space and lay the groundwork for future exploration in unlearning and privacy within the VFL setting, without overstating guarantees beyond what is supported by our analysis.
~~~

---

`Closing Remark`

We appreciate the reviewers’ thoughtful engagement and the constructive feedback that helped refine the clarity of our presentation. The revision incorporates the requested formal definition of process privacy and a small number of clarifications that strengthen the exposition while leaving the core method and results unchanged. These updates address all substantive concerns raised during the review process reflected by one reviewer already increasing their score and another noting that the new formalization resolves their remaining issue.

We hope the improved clarity of the manuscript assists new Area Chair in the final evaluation and contributes meaningfully to advancing ***research on privacy-preserving unlearning in vertical federated learning (VFL) setting***.

---

> ### Author Response · Authors · 2025-11-29
> **Consolidated Reviewer Concerns and Responses (for the New Area Chair’s Reference)**
>
> Following the above, and to support new Area Chair’s assessment, we summarise the key reviewer concerns and present our consolidated responses.
>
> All revisions have been incorporated into the updated manuscript, which remains within the page limit. **These updates are purely editorial**, **refining clarity**, **strengthening narrative coherence**, and **improving readability** without altering the technical content, theoretical guarantees, or empirical results. The core contribution, methodology, and findings remain unchanged from the original submission. The presentation is now more polished and accessible.
>
> ---
>
> **Reviewer  T6TS (Score ↑ 4 → 6) (Confidence 4)**
>
> `Concerns: threat model ambiguity; no formal process privacy; premature leakage analysis; public-data assumption`
>
> ~~~
>
> Response: added a formal definition; reorganized sections so analysis follows the method; clarified what is communicated in the protocol; emphasized the few-shot nature of public anchors and their theoretical and empirical justification.
> ~~~
>
> ---
>
> **Reviewer Dw3V (Score contingent on process privacy) (Confidence 4)**
>
> `Concerns: lack of formal definition on process privacy; feasibility of including such a definition`
>
> ~~~
> Response: added a KL-based formal definition of process privacy; integrated it smoothly into Section 5.4; showed that existing leakage experiments directly instantiate this definition.
> ~~~
>
> ---
>
> **Reviewer XuNo (Maintained 6) (Confidence 3)**
>
> `Concerns: public-data availability in sensitive domains; asynchronous VFL as a common configuration.`
>
> ~~~
> Response: clarified that we only require 15–40 labeled public anchors per class, with theoretical (Theorem 1) and empirical support; highlighted how mixup expands coverage (Fig. 5); clarified that synchronous VFL is standard in current systems and prior vertical unlearning work; discussed asynchronous extensions explicitly as future work in Appendix A.9.
> ~~~
>
> ---
>
> **Reviewer LVms (Score 6; didnt manage to follow-up) (Confidence 3)**
>
> `Concerns: strength of gradient alignment; semi-honest adversary assumption; behavior under sequential deletions.`
>
> ~~~
> Response: clarified the assumptions underpinning Theorem 1 and its scope; restated the semi-honest setting; expanded the limitations and discussion on sequential unlearning behavior in Section 5 and Appendix A.9.
> ~~~

---

### Meta-Review · Area_Chair_W3vh · 2026-01-07

**Summary:**

The reviewers generally agree that the paper addresses a practically relevant and underexplored problem, and that the proposed method is empirically effective and computationally efficient. Reviewers recognized that the approach demonstrates feasible unlearning behavior and reduces label leakage compared to existing baselines.

At the same time, several concerns were raised across reviews. These include:
(1) limited algorithmic and theoretical novelty, as the method primarily adapts existing techniques (manifold mixup, zero-shot class unlearning and gradient-based unlearning) to the VFL setting;
(2) unclear privacy objectives and threat model in the initial submission and the absence of formal privacy guarantees;
(3) assumptions that may limit applicability, such as reliance on a small public labeled dataset, synchronous VFL, etc.

**Reviewer Concerns:**

Concerns that were addressed:

The ambiguity around the threat model and privacy objective was clarified.

Concerns about overstated privacy and performance claims were addressed by tightening the language and clarifying assumptions and scope.

Technical questions regarding mixup hyperparameter sensitivity and sequential unlearning stability were addressed.

Issues related to paper structure and narrative flow were acknowledged and corrected.

Concerns that remain:

The contribution remains incremental in algorithmic and theoretical depth.

Support for stronger practical settings (e.g., no public labeled data, asynchronous VFL, malicious adversaries) remains limited (but may be more suitable as future work).

**Reviewer Scores:**

Reviewer LVms initially gave positive rate, with concerns about robustness under few-shot public data. These concerns were sufficiently addressed through empirical evidence, and the score would likely remain the same or increase slightly.

Reviewer Dw3V raised the strongest initial objections regarding the threat model and privacy definition. After the rebuttal and subsequent clarification, the reviewer indicated that most concerns had been addressed and expressed openness to increasing their rating; their score would likely increase.

Reviewer Dw3V focused on novelty, privacy guarantees, and experimental depth. After clarification and claim refinement, the paper aligns better with the reviewer’s expectations, and the score would likely remain stable or increase marginally (the reviewer expressed in discussion that they are willing to raise their score).

Reviewer XuNo was generally positive, with concerns mainly related to extensions and future robustness. These were appropriately addressed in the rebuttal, and the score would likely remain unchanged.

---

### Decision · Program_Chairs · 2026-01-26

Accept (Poster)